# Field measurements of wake meandering at a utility-scale wind turbine with nacelle-mounted Doppler LiDARs

Peter Brugger[1], Corey Markfort[2], and Fernando Porté-Agel[1]

[1]Wind Engineering and Renewable Energy Laboratory (WiRE), École Polytechnique Fedérale de Lausanne (EPFL), 1015 Lausanne, Switzerland
[2]IIHR-Hydroscience & Engineering, Dept. of Civil and Environmental Engineering, The University of Iowa, Iowa City, IA 52242, USA

**Correspondence:** Peter Brugger (peter.brugger@epfl.ch)

**Abstract.** Wake meandering is a low-frequency oscillation of the entire wind turbine wake that can contribute to power and load fluctuations of downstream turbines in wind farms. Field measurements of two Doppler LiDARs mounted on the nacelle of a utility-scale wind turbine were used to investigate relationships between the inflow and the wake meandering as well as the effect of wake meandering on the temporally averaged wake. A correlation analysis showed a linear relationship between the instantaneous wake position and the lateral velocity that degraded with the evolution of the turbulent wind field during the time of downstream advection. A low-pass filter proportional to the advection time delay is recommended to remove small scales that become decorrelated even for distances within the typical spacing of wind turbine rows in a wind farm. The results also showed that the velocity at which wake meandering is transported downstream was slower than the inflow wind speed, but faster than the velocity at the wake center. This indicates that the modelling assumption of the wake as an passive scalar should be revised in the context of the downstream advection. Further, the strength of wake meandering increased linearly with the turbulence intensity of the lateral velocity and with the downstream distance. Wake meandering reduced the maximum velocity deficit of the temporally averaged wake and increased its width. Both effects scaled with the wake meandering strength. Lastly, we found that the fraction of the wake turbulence intensity that was caused by wake meandering decreased with downstream distance contrary to the wake meandering strength.

## 1 Introduction

The wind turbine wake is a flow region of reduced wind speed and increased turbulence that extends downstream of a wind turbine for several rotor diameters. In wind farms the wake of an upstream turbine becomes the inflow of a downstream turbine, leading to power losses and increased mechanical wear, which translates into an increased cost of the energy. Therefore, understanding and predicting characteristics of wind turbine wakes has received extensive attention in the literature (see reviews by Vermeer et al. (2003); Sørensen (2011); Sanderse et al. (2011); Mehta et al. (2014); Stevens and Meneveau (2017); Porté-Agel et al. (2020)).

One characteristic of wind turbine wakes is wake meandering, a low frequency, large scale oscillation of the entire wake in the lateral and vertical direction (Taylor et al., 1985). Two theories have been presented as the origin of wake meandering:

(i) a passive advection of the entire wake by large scale turbulence of the inflow (Larsen et al., 2008), and (ii) an intrinsic shear instability of the wake characterized by periodic vortex shedding (Medici and Alfredsson, 2006). Support for the passive advection hypothesis has been presented in Trujillo et al. (2011); Keck et al. (2014) and for the shear instability hypothesis in Medici and Alfredsson (2006); Heisel et al. (2018); Yang and Sotiropoulos (2019). The passive advection hypothesis forms the basis of the dynamic wake meandering model (Larsen et al., 2008), which assumes that wake meandering is driven by large scale turbulence (with two rotor diameters used as a threshold). While the passive advection hypothesis assumes the inflow wind speed as the downstream propagation velocity of the wake meandering, Bingöl et al. (2010) reported better agreement between the dynamic wake meandering model and field measurements using an reduced wake velocity from the Jensen (1983) wake model.

Several characteristics of wake meandering have been reported in literature mainly from wind-tunnel experiments. The strength (or amplitude) of wake meandering is larger in the lateral direction than in the vertical direction (España et al., 2012; Bastankhah and Porté-Agel, 2017), increases with downstream distance (Garcia et al., 2017), and depends on incoming boundary-layer characteristics (Bastankhah and Porté-Agel, 2017). The instantaneous horizontal wake position is correlated to the upstream transverse velocity for large wavelengths and the correlation decreases with downstream distance (Muller et al., 2015; Aubrun et al., 2015).

In this paper, we use field measurements at a utility scale wind turbine across a wide range of turbulence intensities and wind speeds to investigate (i) the effect of the inflow state on the correlation between lateral velocity and instantaneous wake position, (ii) the downstream advection velocity of wake meandering, and (iii) the effect of wake meandering on the temporal averaged velocity deficit and the turbulence intensity of the wake. The present study extends on the investigations of Trujillo et al. (2011) on the effect of wake meandering on the mean wake and the turbulence intensity to a wider range of atmospheric conditions, validates the findings of Muller et al. (2015) from wind tunnel scale turbines with field experiments at a utility scale wind turbine, and presents new insights into the limits of passive advection based wake meandering predictions.

## 2 Methods

This section introduces the measurement site, the measurement instruments, and the analysis approach of the measured data.

### 2.1 Measurement site and Doppler LiDAR setup

The measurement campaign was conducted at an isolated 2.5 MW Liberty C96 wind turbine from Clipper Windpower with a hub height ($z_{hub}$) of 79 m and a rotor diameter ($D$) of 96 m located at the Kirkwood Community College campus in Cedar Rapids, Iowa, United States (Fig. 1). The immediate surroundings of the wind turbine as well as the area to the north and west are urbanized. The area to the south and east is agricultural farm land. The topography is characterized by gentle rolling hills. The data of the supervisory control and data acquisition (SCADA) system of the wind turbine is available providing the temporal mean values of the wind speed at hub height ($\overline{u}_{hub}$, where the bar indicates a temporal average), the nacelle position, and the yaw misalignment of the nacelle.

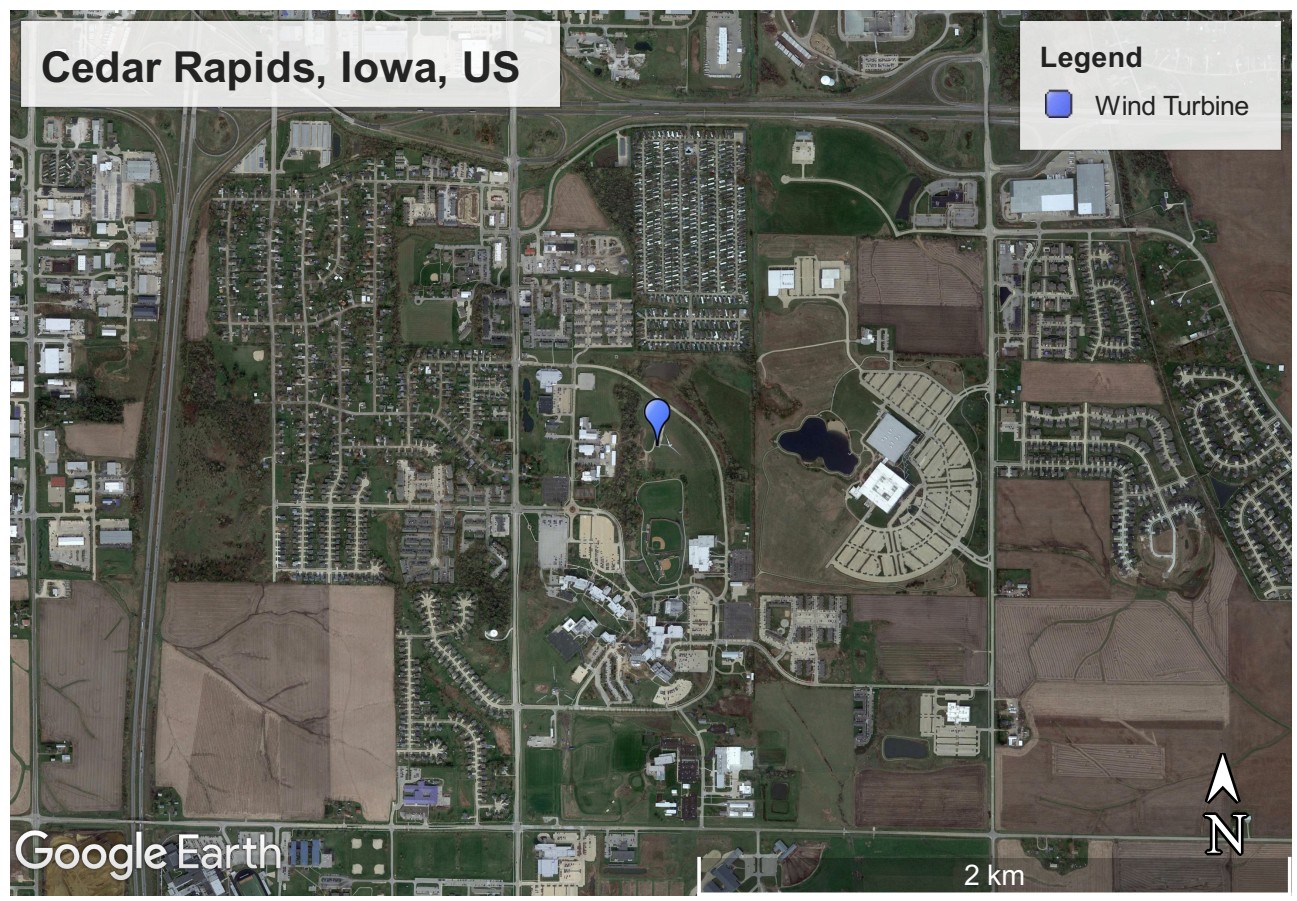

**Figure 1.** Satellite image of the measurement site with the location of the wind turbine (© Google Earth). The wind turbine coordinates are $41.9165°$ latitude and $-91.6508°$ longitude.

Two Doppler LiDARs of the type Stream Line manufactured by Halo Photonics were installed on the nacelle of the wind turbine. The Doppler LiDAR measures the radial (or line-of-sight) velocity along a laser beam that is emitted from a movable scanner head. The Doppler LiDARs were configured to measure with a sampling frequency of 3 Hz and a range gate length of 18 m. A positive value of the radial velocity corresponds to a motion away from the Doppler LiDAR, and a negative radial velocity is a motion towards the Doppler LiDAR.

The backward facing Doppler LiDAR was programmed to perform 230 successive Plan Position Indicator (PPI) scans of the wind turbine wake at hub height covering an azimuth range of $\pm12°$ from the rotor axis (Fig. 2a, red). The scanner was starting at $az = 168°$ and travelling at a speed of $6°\ \mathrm{s}^{-1}$ to $az = 192°$ while continuously measuring, which resulted in an azimuth resolution of $2°$ (Fig. 2b). This scan pattern was completed within a 29 minute period (a single PPI with return to the starting

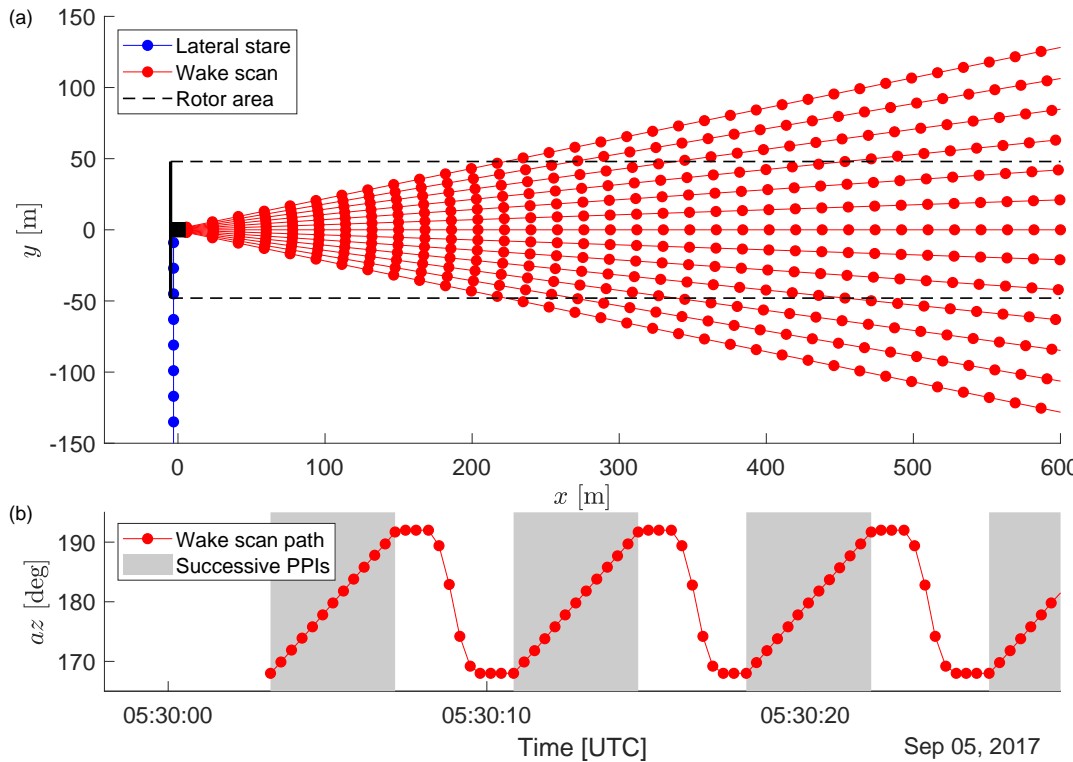

**Figure 2.** Scan patterns of the nacelle mounted Doppler LiDARs viewed from top (a). Wake scans of the backward facing Doppler LiDAR (red) were accompanied by measurements in a lateral staring mode of the forward facing Doppler LiDAR (blue). LiDAR beams are shown as lines with range gate centers indicated as points. The wind turbine is stylized in black (not to scale) and rotor area is indicated with black dashed lines. The bottom panel shows the scanner path for a section of a wake scan (b), where the grey area indicates the successive PPIs that together become a wake scan.

position took $7.2 - 7.6$ s). A full set of 230 PPI scans will be defined as "wake scan" in the following. These wake scans were scheduled every second hour.

Coinciding with a wake scan, the forward facing Doppler LiDAR was measuring in a horizontal staring mode at a $90°$ angle to the rotor axis for 14 min at a temporal resolution of 3 Hz (Fig. 2, blue). The rejection criteria for wake scans not suited for further analysis based on data quality, turbine yaw activity, and inflow characteristics will be presented at the beginning of Sect. 3.

## 2.2 Post-processing of measurments from the wake scanning Doppler LiDAR

The wake scans are processed along the following steps to obtain the instantaneous wake position:

1. Doppler LiDAR measurements with a signal-to-noise ratio (SNR) of less than $-17$ dB are rejected (Pearson et al., 2009).

2. The remaining radial velocities are gridded on a polar coordinate system $u_r(\phi, r, t)$ with an angular ($\phi$) resolution of $2°$, a radial ($r$) resolution of $18$ m, and a time stamp ($t$) aligning with the PPIs of the wake scans. The $az$ positions of the LiDAR scans and the $\phi$ positions of the polar coordinate system can have a difference of $0.2°$ towards the end of a PPI resulting from small variations of the scanner behaviour and fluctuations of the measurement frequency from PPI to PPI. Multiple measurements are available for the outside grid points due to a short resting time of the scanner at the turn-around point and the measurements closest in time are used at those grid points.

3. The transition to a Cartesian coordinate system is made with $y = r\sin^{-1}(\phi)$ and approximating $x = r\cos^{-1}(\phi)$ with $x = r$ (spatial error $< 3\%$ based on geometry).

4. A instantaneous velocity deficit is computed with

$$\Delta u_r(x, y, t) = \max_y(u_r(x, y, t)) - u_r(x, y, t), \tag{1}$$

where $\max_y(u_r(x, y, t))$ is the maximum of the velocity observed for each PPI of the wake scan at a given downstream distance. While this is not the longitudinal velocity deficit typically used in the literature, the effect of the lateral and vertical velocity on the wake center position is negligible as shown in Appendix A. Appendix B explains our reasoning to compute the instantaneous velocity deficit relative to the instantaneous velocity outside of the wake and not relative to the mean wind speed at hub height.

5. The instantaneous position of the wake center is detected in analogy to the center-of-mass from the velocity deficit with

$$y_{com}(x, t) = \frac{\sum_y y \Delta u_r(x, y, t)}{\sum_y \Delta u_r(x, y, t)}. \tag{2}$$

The above processing steps are applied for downstream distances between $xD^{-1} = 4$ and $xD^{-1} = 9$. The double-peak shape of the near-wake was a problem for the detection of $y_{com}$ for $xD^{-1} < 4$ and the decline of the SNR with increasing range led to gaps in the data for $xD^{-1} > 9$.

## 2.3 Post-processing of measurements from the lateral staring Doppler LiDAR

The time series of the lateral velocity component, $v(t)$, is obtained from the 7th range gate at $y = 117$ m of the Doppler LiDAR operating in the lateral staring mode. Range gates closer than $y = 117$ m were either affected by near range problems of the Doppler LiDAR (the first four range gates), or $v(t)$ was biased towards motions away from the wind turbine and higher standard deviations compared to greater distances (5th and 6th range gate). The latter might be caused by the influence of the wind turbine on the wind field. We inverted the sign of the Doppler velocity in order to have positive velocities towards the positive $y$-direction in Fig. 2. We assume horizontal homogeneity of the inflow and that $v(t)$ is representative for the lateral inflow velocity over the rotor area. Measurements with a SNR below $-17$ dB are removed from the time series and the gaps are interpolated linear.

Further, we derived two quantitative measures to characterize the inflow state from the lateral staring Doppler LiDAR. First, the lateral turbulence intensity is given by

$$I_v = \sigma(v)\overline{u}_{hub}^{-1}, \tag{3}$$

where $\sigma(v)$ is the standard deviation of $v(t)$ over the 14-minute period. It quantifies the strength of the lateral velocity fluctuations relative to the mean wind speed. Second, the integral time scale of the lateral velocity component, $T_{i,v}$, is estimated from the noise corrected auto-correlation function of $v(t)$ by fitting an exponential decay law (Lothon et al., 2006). It is a measure for the correlation length and can be interpreted as the scale of the dominant eddies of the turbulent wind field.

## 2.4 Advection velocity

Advection velocity is referring to the velocity of downstream propagation of wake meandering and is defined as

$$u_{adv} = \frac{\Delta x}{\Delta T_{adv}}, \tag{4}$$

where $\Delta x$ is a spatial separation in the $x$-direction between two measuring points and $\Delta T_{adv}$ is the time delay between the wake meandering signals at the two points. $\Delta x$ is known from the scan geometry and $\Delta T_{adv}$ is determined from the time lag of the maximized cross-correlation as described in the following steps. The cross-correlation is computed between $v$ and $y_{com}$ as well as between $y_{com}$ at two spatially separated downstream distances. The terms upstream signal and downstream signal will refer to their relative streamwise position to each other.

1. Both, the upstream signal ($v$ or $y_{com}(x - 0.5\Delta x)$) and the downstream signal ($y_{com}(\Delta x)$ or $y_{com}(x + 0.5\Delta x)$), are low-pass filtered with a moving mean that has a window length of $\Delta x \overline{u}_{hub}^{-1}\beta^{-1}$ with $\beta = 3$ (Cheng and Porté-Agel, 2018). The inverse proportionality of $\beta$ to $I_v$ proposed by Corrsin (1963) is not used here, because it is only valid for a convective atmospheric boundary layer, which cannot be ensured for all wake scans in our data set (e.g. at night-time).

2. If the upstream signal has a higher temporal resolution in the case of $v(t)$, each time step of the downstream signal is paired with the temporally closest time step of the upstream signal for synchronization.

3. The cross-correlation function between the upstream signal and the downstream signal is computed for time lags between $0.1\Delta x \overline{u}_{hub}^{-1}$ and $1.5\Delta x \overline{u}_{hub}^{-1}$.

4. If the cross-correlation function has a local maximum (i.e. not equal to the smallest or largest lag) with a correlation above 0.5, the corresponding time lag is selected as $\Delta T_{adv}$ and $u_{adv}$ is computed with Eq. (4).

The time series of $y_{com}$ has a temporal resolution of $\Delta t_s = 7.2$ s due to the time a single PPI scan takes (Sect. 2.1). Therefore, $\Delta T_{adv}$ has to be a multiple of $\Delta t_s$, which results in a loss of precision ($\epsilon$) for $u_{adv}$ given by

$$u_{adv} \pm \epsilon = \frac{\Delta x}{\Delta T_{adv} \mp 0.5\Delta t_s}. \tag{5}$$

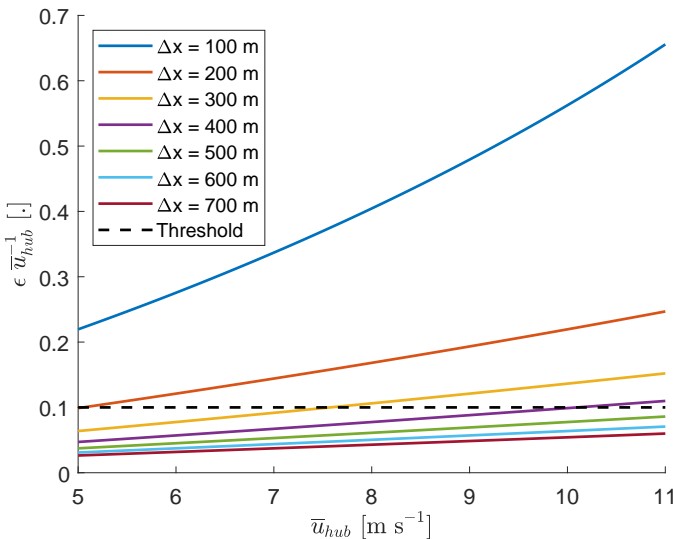

**Figure 3.** Error of the estimated downstream advection velocity of wake meandering ($\epsilon$) as a function of the mean wind speed at hub height ($\overline{u}_{hub}$) and the spatial separation ($\Delta x$). If the error becomes larger than the threshold, the detected advection velocity is rejected.

Assuming an upper limit of $u_{adv} = \overline{u}_{hub}$ (Sect. 3.1.2 will show that $u_{adv}$ is smaller or equal to $\overline{u}_{hub}$ for all cases), it is shown in Fig. 3 that the precision decreases for high wind speeds or small spatial separations. Only combinations of wind speed and spatial separation that result in a precision of at least $0.1\overline{u}_{hub}$ will be used.

## 3   Results

The scan setup described in Sect. 2.1 was implemented between 19 August 2017 and 2 October 2017. Data before 5 September 2017 is discarded due to a time synchronisation problem with one of the Doppler LiDARs. To ensure high quality measurement data and suitable conditions for the investigation, wake scans were rejected if

- low SNR of the wake scanning Doppler LiDAR led to gaps in the measurement data, which is quantified by a rejection rate of more than 0.5% at any range gate between $xD^{-1} = 4$ and $xD^{-1} = 9$ (Sect. 2.2);

- the SCADA data reported a non-operational wind turbine, yaw movements of the nacelle, or a mean wind speed below $5$ m s$^{-1}$;

- the wake was partially outside of the Doppler LiDAR field-of-view, quantified by more than 25% of either $\Delta u_r(6D, x\sin(168°), t)$ or $\Delta u_r(6D, x\sin(192°), t)$ being larger than the temporal mean of $\frac{1}{2}\max_y(\Delta u_r(6D, y, t))$.

The remaining data set consists of 43 wake scans accompanied by measurements of the lateral velocity component of the

inflow. A characterization of the data set is shown in Fig 4. The data set covers a wind speed range between $5$ m s$^{-1}$ and $11$ m s$^{-1}$ (Fig 4a), and lateral turbulence intensities between $< 1\%$ and $8\%$ (Fig 4b). The check on the wake position within

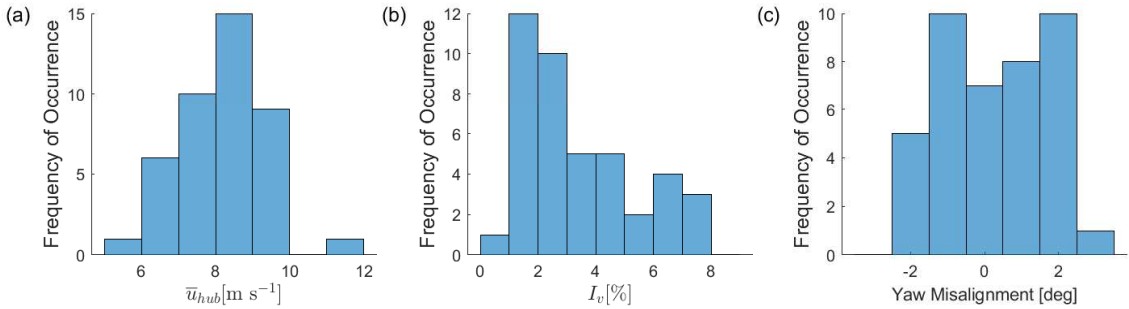

**Figure 4.** Distribution of the mean wind speed at hub height from the SCADA data (a), the lateral turbulence intensity from the lateral staring Doppler LiDAR (b), and the mean yaw misalignment of the wind turbine from the SCADA data (c).

the wake scanning Doppler LiDAR's field-of-view implicitly ensured a good alignment of the wind turbine with the mean wind direction (Fig 4c).

The results presented in the subsequent subsections are not differentiated according to the wind direction of the inflow. We assume that individual buildings in the vicinity of the wind turbine should not affect the measurements at $z_{hub}$ based on the blending height concept. Note that the roughness sublayer in an urban environment should not extend further than five times the building height (Grimmond and Oke, 1999), which is lower than $z_{hub}$ for an assumed building height of 10 m (two-story and three-story buildings).

### 3.1 Wake meandering

The first part of the results investigates the relationships between the instantaneous lateral velocity of the inflow and the instantaneous wake position.

### 3.1.1 Correlation between lateral velocity and wake position

Assuming the wake is advected passively and the turbulence field does not evolve during the downstream advection, a linear relationship between the lateral velocity at the turbine location and the wake center position at a given downstream distance would be expected. For an evolving turbulent wind field, the autocorrelation function of a variable is expected to decay exponentially for idealized isotropic and homogeneous turbulence (Von Kármán, 1948). The rate at which the autocorrelation function decays is described by the integral time scale. Hence, we would expect a lower correlation between $v(t)$ and $y_{com}(t)$ for a given time lag (here the time delay due to advection), if the dominant features of the wind field have a short lifespan, which is equivalent to a fast decay of the autocorrelation function and a short integral time scale. Vice versa, a higher correlation is expected for a long integral time scale.

This hypothesis is confirmed in Fig. 5a, where the abscissa $T_{i,v}\Delta T^{-1}$ quantifies aforementioned interplay of the time delay and the evolution of the turbulent field. $T_{i,v}$ is the integral time scale of the lateral velocity component (Sect. 2.3) and $\Delta T$ is the time delay due to downstream advection as defined in the figure legend. Low values of the correlation between $v(t)$ and $y_{com}(t)$

are observed if the time delay is longer than the lifespan of the dominant eddies and the correlation increases, if the lifespan of the dominant eddies increases relative to the time delay. This result holds for the assumption of a downstream advection with the mean wind speed (black crosses in Fig. 5) as well as for the subset of the data set with a successful detection of the avection velocity based on the maximized cross-correlation (blue crosses in Fig. 5). The findings agree with expectation that the correlation decreases, if a relatively larger amount of small scale turbulence is included that is not expected to be correlated. Figure 6 shows an example case for a wake scan with high correlation (Fig. 6a and 6c) and a wake scan with low correlation (Fig. 6b and 6d).

Larsen et al. (2008) hypothesised that wake meandering is driven by large scale turbulence and recommended $2D$ as a low-pass filter threshold. Applying a low-pass filter equivalent to $2D$ to $v(t)$ and $y_{com}(t)$ does increase the correlation compared to the unfiltered data (Fig. 5b) supporting the assumption of large scale turbulence as the driver of wake meandering.

Cheng and Porté-Agel (2018) proposed a low-pass filter threshold proportional to the time delay due to downstream advection based on Taylor's diffusion theory (Taylor, 1922). Their filter size is given by $x\overline{u}_{hub}^{-1}\beta^{-1}$ with $\beta = 3$ to account for the difference between Lagrangian and Eulerian scales (Angell et al., 1971). The results for this low-pass filter threshold are similar to the ones for the threshold of $2D$ at $xD^{-1} = 6$ (Fig. 5c), but for $xD^{-1} > 6$ this filter threshold maintains an improvement of the correlation of approximately $0.2$, while the $2D$ threshold decreases with downstream distance (Fig. 5d). This shows that the evolution of the turbulence field becomes important at sufficiently large downstream distances and removing scales which are not expected to be correlated improves the correlation between $v(t)$ and $y_{com}(t)$.

Based on the above findings we recommend to use a low-pass filter threshold based on the advection time delay as suggested by Cheng and Porté-Agel (2018) with a lower limit equivalent to $2D$ for wake meandering predictions. This accounts for large scale turbulence as the origin of wake meandering at short downstream distances, but also accounts for the evolution of the turbulent wind field once it becomes relevant at larger downstream distances. The results also show that the evolution of the turbulent wind field frequently becomes relevant on scales that are similar to the distance between wind turbine rows in a wind farm.

### 3.1.2  Wake meandering strength and lateral turbulence intensity

The relationship between the lateral turbulence intensity of the inflow and the wake meandering strength is investigated. It can be visually observed that the fluctuations of $y_{com}$ increase with the turbulence intensity of the lateral velocity component (Fig. 7). Quantifying the strength of wake meandering as the temporal standard deviation of $y_{com}$, an increase of the wake meandering strength with the lateral turbulence intensity is observed (Fig. 8a). The wake meandering strength also increases with downstream distance (Fig. 8b), which is explained with the longer downstream advection time leading to larger lateral displacement of the wake based on the lateral velocity. The range of the wake meandering strength at a given downstream distance in Fig. 8b is explained by the turbulence intensity range of the data set.

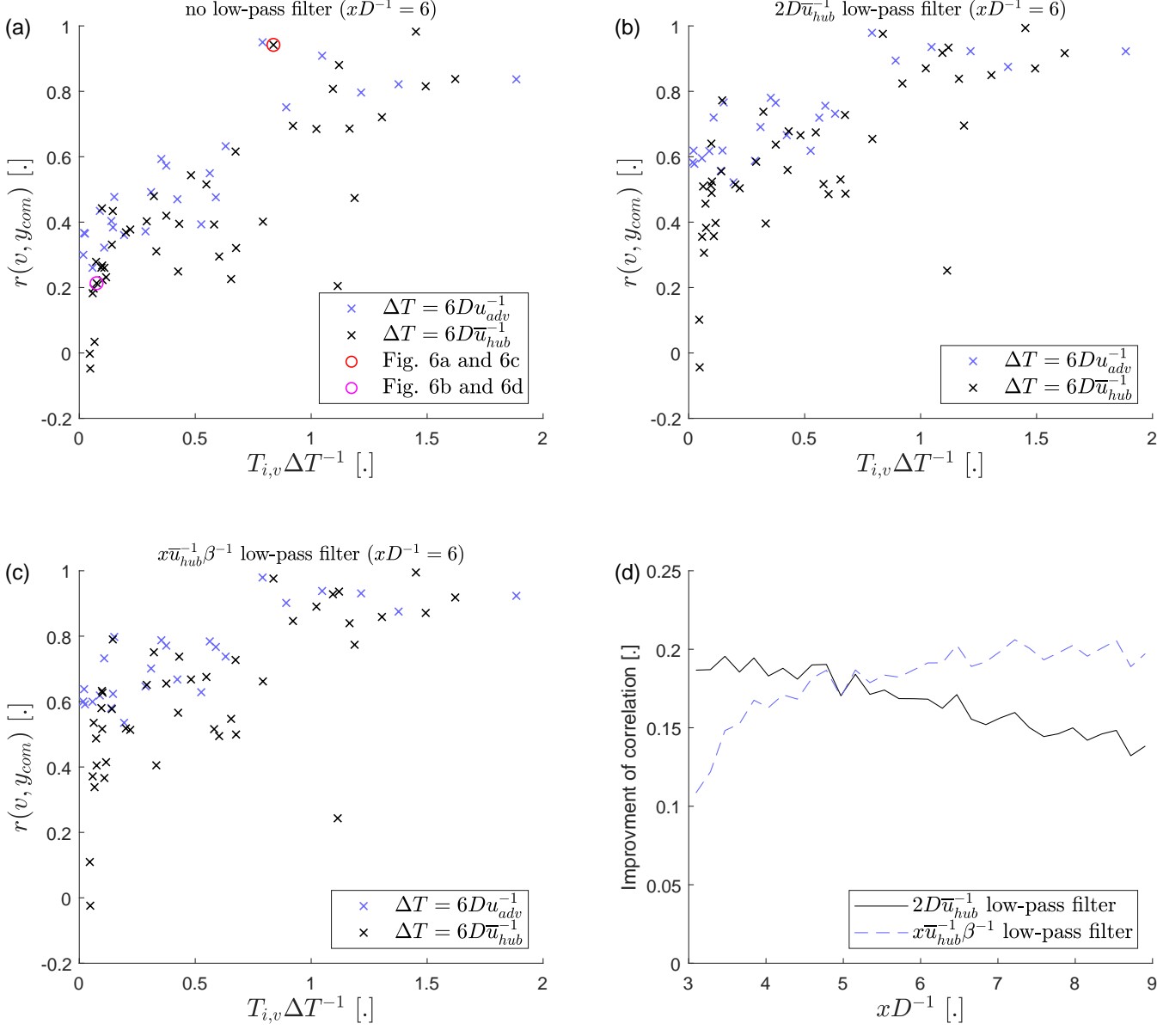

**Figure 5.** The correlation coefficient between the time-shifted lateral velocity and the wake center position ($r(v, y_{com})$) without low-pass filtering (a), with low-pass filtering by a moving mean with a window width of $2D\overline{u}_{hub}^{-1}$ (b), and with low-pass filtering by a moving mean with a window width of $x\overline{u}_{hub}^{-1}\beta^{-1}$ (c). Data points using a time delay based on the mean wind speed are shown with black crosses and data points using a time delay based on the maximized cross-correlation are shown with blue crosses (see Sect. 2.4). The bottom-right panel (d) shows the ensemble-averaged improvement of the correlation with low-pass filtering compared to the unfiltered correlation coefficient as a function of the downstream distance.

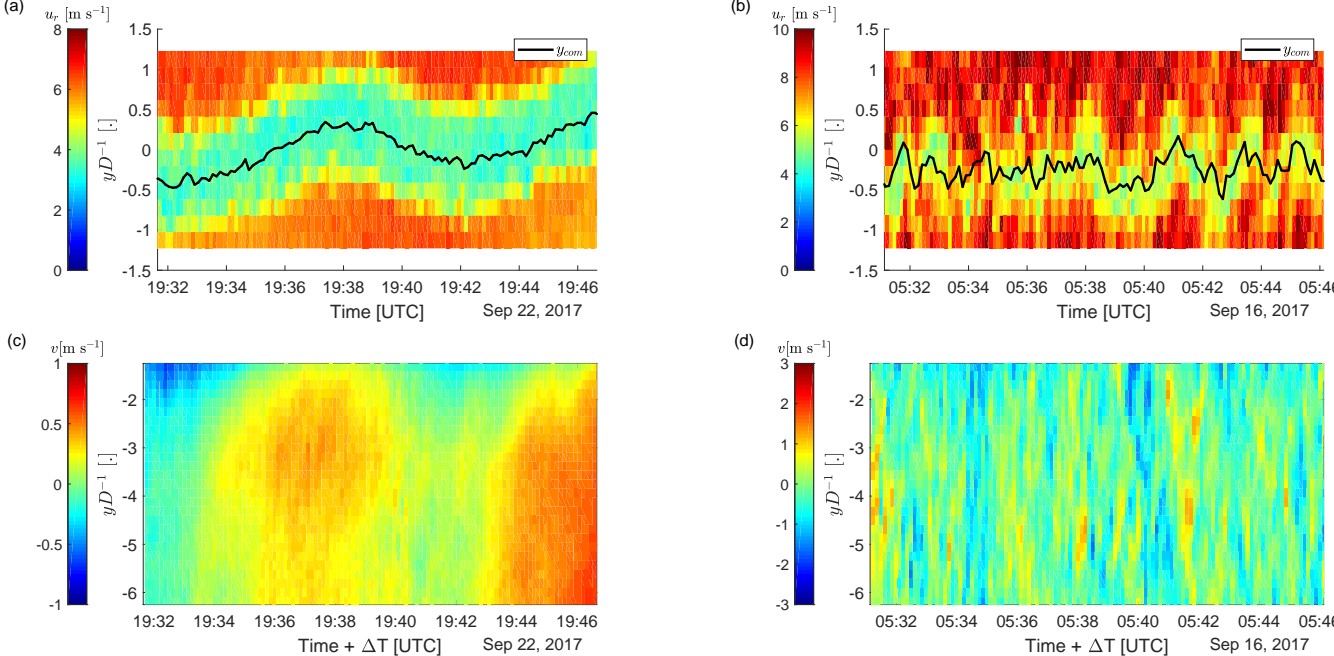

**Figure 6.** The time series of the line-of-sight velocity profiles at $xD^{-1} = 6$ from the wake scanning Doppler LiDAR (a,b) and the time series of the lateral velocity along the laser beam of the lateral staring Doppler LiDAR (c,d). The example case in the left column (a,c) was chosen for its visually clear relationship between lateral velocity and wake center position. The example case in the right column (b,d) illustrates the absence of a correlation between lateral velocity and wake center position even though wake meandering is visible. The lateral velocity was time shifted with $\Delta T = 6Du_{adv}^{-1}$ for (c), and $\Delta T = 6D\overline{u}_{hub}^{-1}$ for (d) to account for a time delay arising due to the downstream advection.

### 3.1.3 Downstream advection velocity of wake meandering

The advection velocity of wake meandering is investigated based on the time delay determined with a cross-correlation approach (Sect. 2.4). The majority of downstream advection velocities found from the time delay between $v$ and $y_{com}$ range between $0.6\overline{u}_{hub}$ and $1.0\overline{u}_{hub}$ (Fig. 9a). The majority of advection velocities found from cross-correlation of $y_{com}$ at two downstream distances showed values between $0.7\overline{u}_{hub}$ and $0.9\overline{u}_{hub}$ (Fig. 9b). For both results, only cases with a sufficiently high cross-correlation and a distinct peak of the cross-correlation function were considered (Sect. 2.4). We assume that the results of the advection velocity reported in Fig. 9b are more robust compared to Fig. 9a, because the underlying correlations are higher, no assumptions on the origin of wake meandering are made, and any time synchronisation issues between the two Doppler LiDARs cannot affect the result. The found advection velocities are in most cases lower then $\overline{u}_{hub}$, but higher than the velocity at the wake center (Fig. 9c). This finding is in line with Bingöl et al. (2010), who reported smaller errors of the dynamic wake meandering model if a reduced downstream advection velocity was used. The results are also in agreement with Zong and Porté-Agel (2020), who showed analytically that the advection velocity is bounded between $0.5\overline{u}_{hub}$ and $\overline{u}_{hub}$.

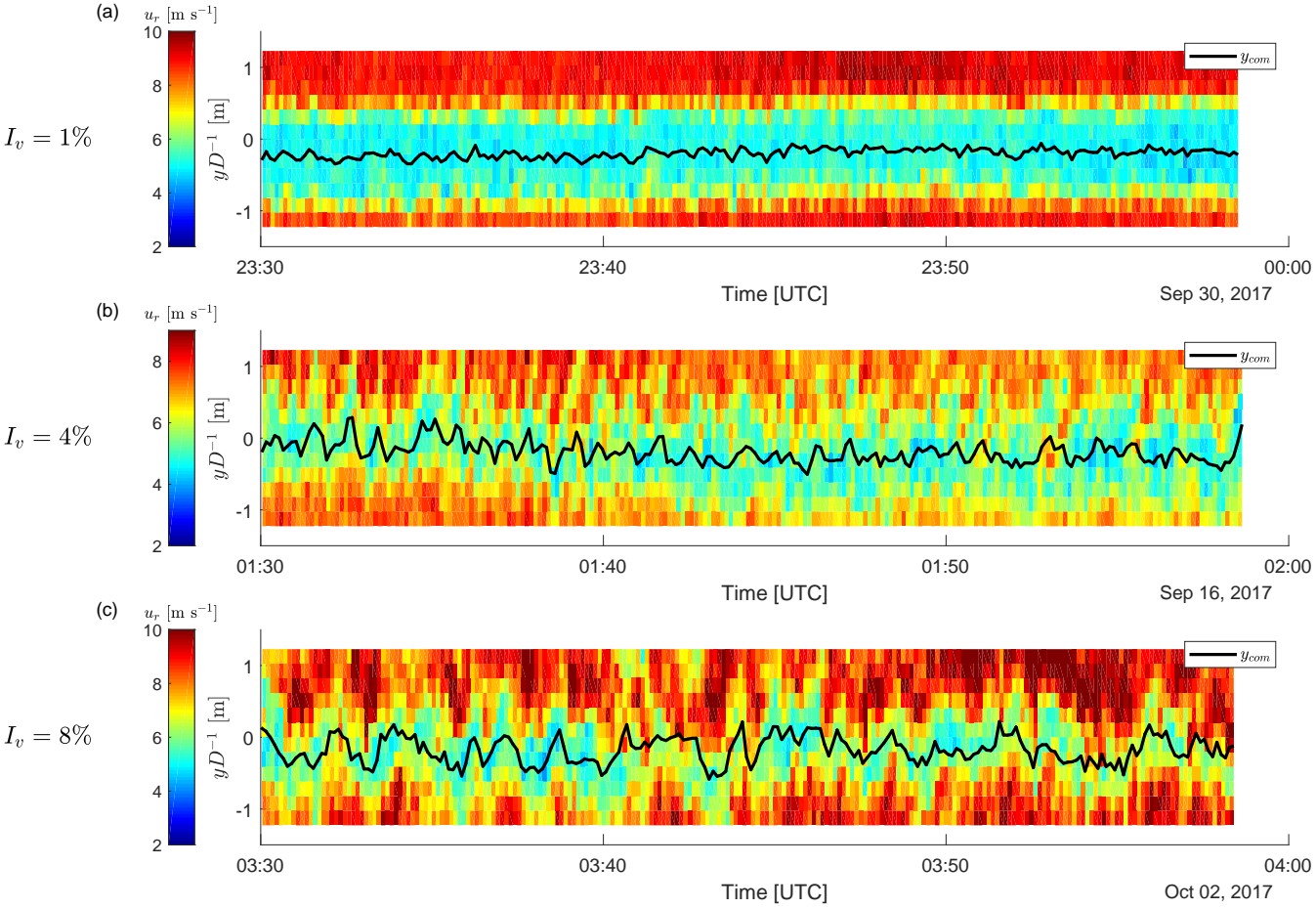

**Figure 7.** Time series of the radial velocity profiles from the wake scanning Doppler LiDAR at $xD^{-1} = 6$ for low (a), medium (b), and high (c) lateral turbulence intensity ($I_v$). The three example cases were chosen to span the range of turbulence intensities present within the data set.

Cheng and Porté-Agel (2018) recommended the average of the mean wind speed and the velocity at the wake center as an estimate for the advection velocity. It is given by $\overline{u_a}(x) = 0.5(\overline{u}_{hub} + \overline{u}_{cen}(x))$ where $\overline{u}_{cen}(x)$ is the temporal average of $u_r(x, y_{com}, t)$. A comparison of the detected advection velocities from the cross-correlation approach and $\overline{u_a}$ shows reasonable agreement at a correlation of 0.7 (Fig. 10).

### 3.2 Effect on the time averaged wake

The second part of the results is investigating the effect of the wake meandering on the properties of the time-averaged wake. The effect of wake meandering is investigated by comparing the wake in the nacelle frame of reference and the meandering frame of reference following the apporach of Trujillo et al. (2011). The transformation from the nacelle frame of reference to

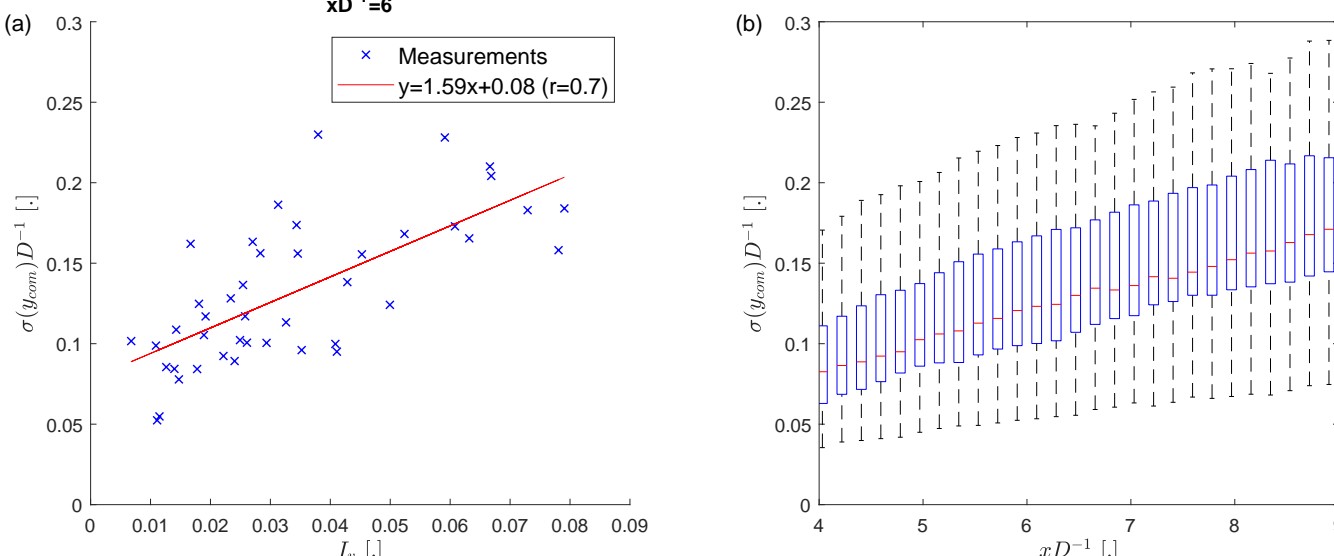

**Figure 8.** The normalized wake meandering strength ($\sigma(y_{com})D^{-1}$) as a function of the lateral turbulence intensity ($I_v$) at $xD^{-1} = 6$ (a). The legend shows the equation of a linear fit to the measurements (red line) and their correlation coefficient. The distribution of the observed wake meandering strength as a function of downstream distance (b). The whiskers show the range of the data, the top and bottom of the blue box indicate the 25th and 75th percentile, and the red center marker is the median.

the meandering frame of reference is given by

$$\widetilde{y} = y - y_{com}, \tag{6}$$

where the tilde is indicating the meandering frame of reference. After the transformation, the measurement data in the meandering frame of reference is interpolated on a regular grid using the nearest available measurement value for each grid point in the lateral direction. An example of the transformation is shown in Fig. 11a and 11b. This method of transformation retains fluctuations of the wake center position smaller than the azimuth resolution of the wake scans. The irregular edge of the meandering frame of reference is caused by the limited azimuth range of the wake scans.

### 3.2.1 Mean Velocity Deficit

First, the effect of wake meandering on the longitudinal mean velocity deficit is investigated. It is given by

$$\overline{\Delta u}(x,y) = \frac{\overline{\Delta u_r}(x,y)}{\cos(\phi - 180°)} \tag{7}$$

for the nacelle frame of reference, and by

$$\overline{\Delta u}(x,\widetilde{y}) = \frac{\overline{\Delta u_r}(x,\widetilde{y})}{\cos(\phi - \sin^{-1}(y_{com}/x) - 180°)} \tag{8}$$

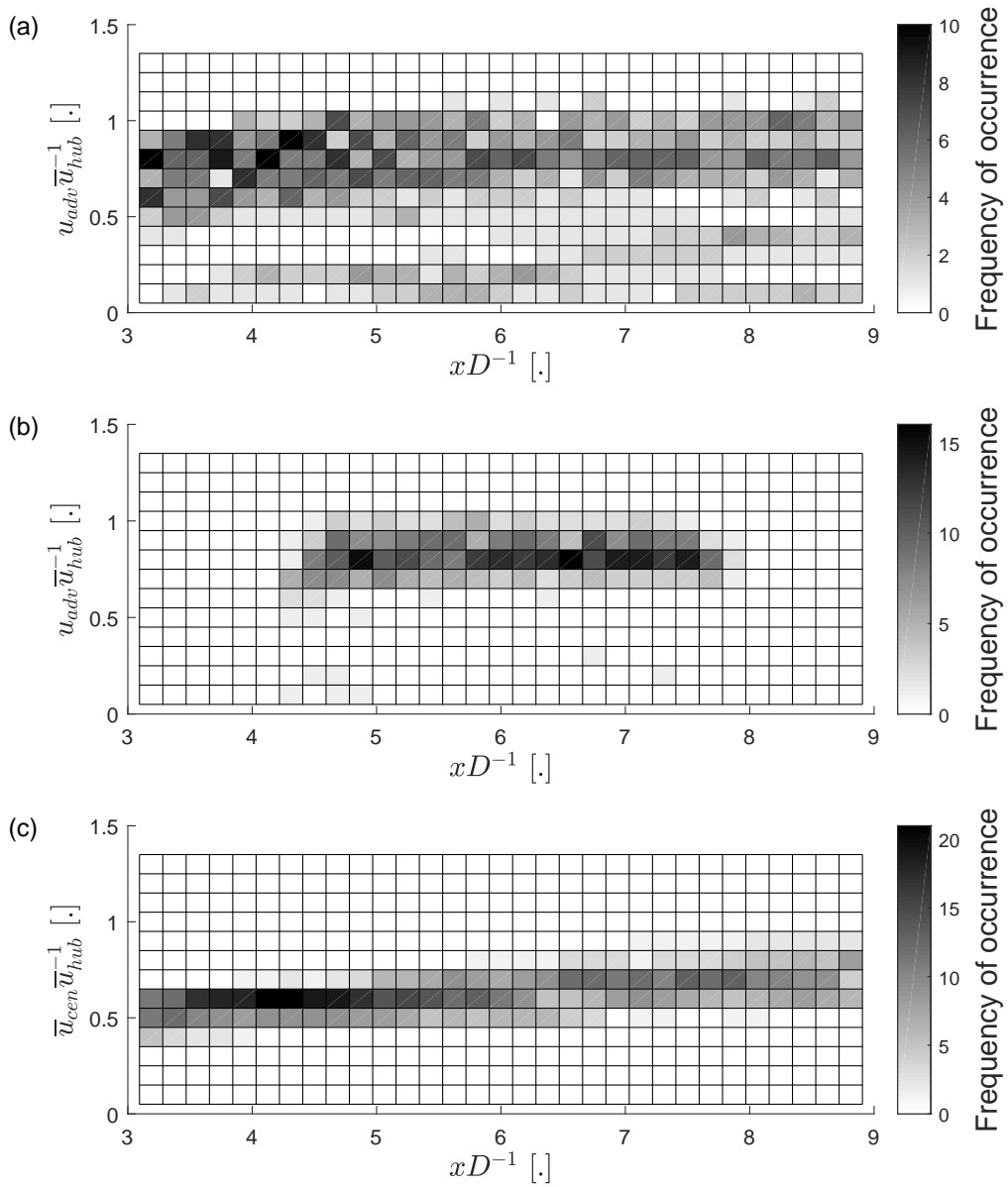

**Figure 9.** Frequency of occurrence of the advection velocity based on the cross-correlation between $v(t)$ and $y_{com}(t)$ (a), the advection velocity based on the cross-correlation between $y_{com}(t)$ at two different downstream distances (b), and the wake center velocity based on the temporal average of $u_r(x, y_{com}, t)$ (c) normalized by $\overline{u}_{hub}$. The absence of data in (b) for $xD^{-1} < 4.5$ and $xD^{-1} > 7.5$ is caused by the error threshold for $u_{adv}$ (Fig. 3).

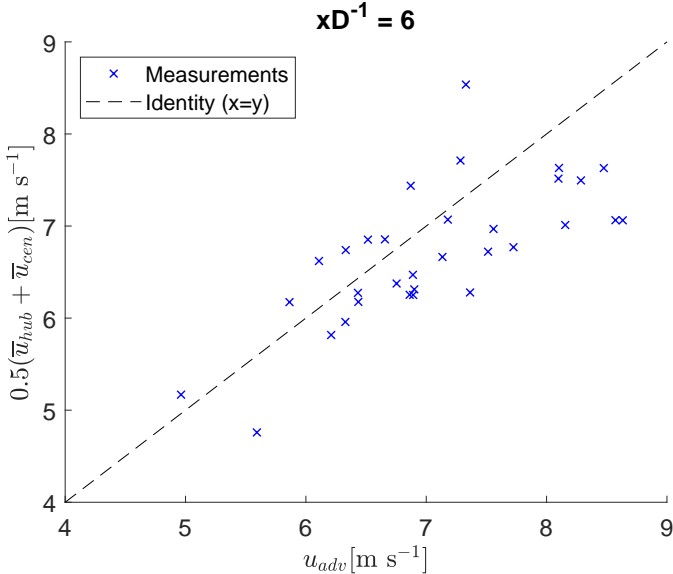

**Figure 10.** Comparison of the observed advection velocities with an estimation based on the average of the mean wind speed and the velocity at the wake center.

for the meandering frame of reference, where a bar indicates a temporal average. A qualitative comparison of the mean velocity deficit profile in both frames of reference shows a slightly deeper and narrower wake in the meandering frame of reference (Fig. 11c). For a quantitative investigation of the full data set, a Gaussian function given by

$$f(y) = C \exp\left(\frac{(y - y_0)^2}{4\sigma_y^2}\right) \tag{9}$$

is fitted to the mean velocity deficit profile in both frames of reference. The fit coefficients $\sigma_y$ and $C$ describe the wake width and the wake depth, respectively. To assure that only cases with an Gaussian velocity deficit are considered, a correlation coefficient of at least $0.99$ between fit and measurements is required or the result is discarded.

The results show that the depth of the mean wake in the nacelle frame of reference is smaller compared to the meandering
frame of reference and the difference increases with wake meandering strength (Fig. 12a). At the same time the wake is wider in the nacelle frame of reference and the effect on the width increases with the wake meandering strength (Fig. 12b). Both observations agree with modeling results presented in Braunbehrens and Segalini (2019) and their explanation of a quasi-steady velocity deficit of the instantaneous wake that is spatially shifted and ensemble averaged to yield the time averaged wake.

As the wake meandering strength increases with downstream distance (Fig. 8b), it could be expected that the observed differences of $C$ and $\sigma_y$ between the two frames of references also increase with downstream distance. For $C$ this increase is not clearly observed (Fig. 12c), but $\sigma_y$ shows the expected increase with downstream distance (Fig. 12d). It is possible that the expected trend for $C$ is masked by the measurement errors, because its amplitude decreases with $xD^{-1}$ due to the

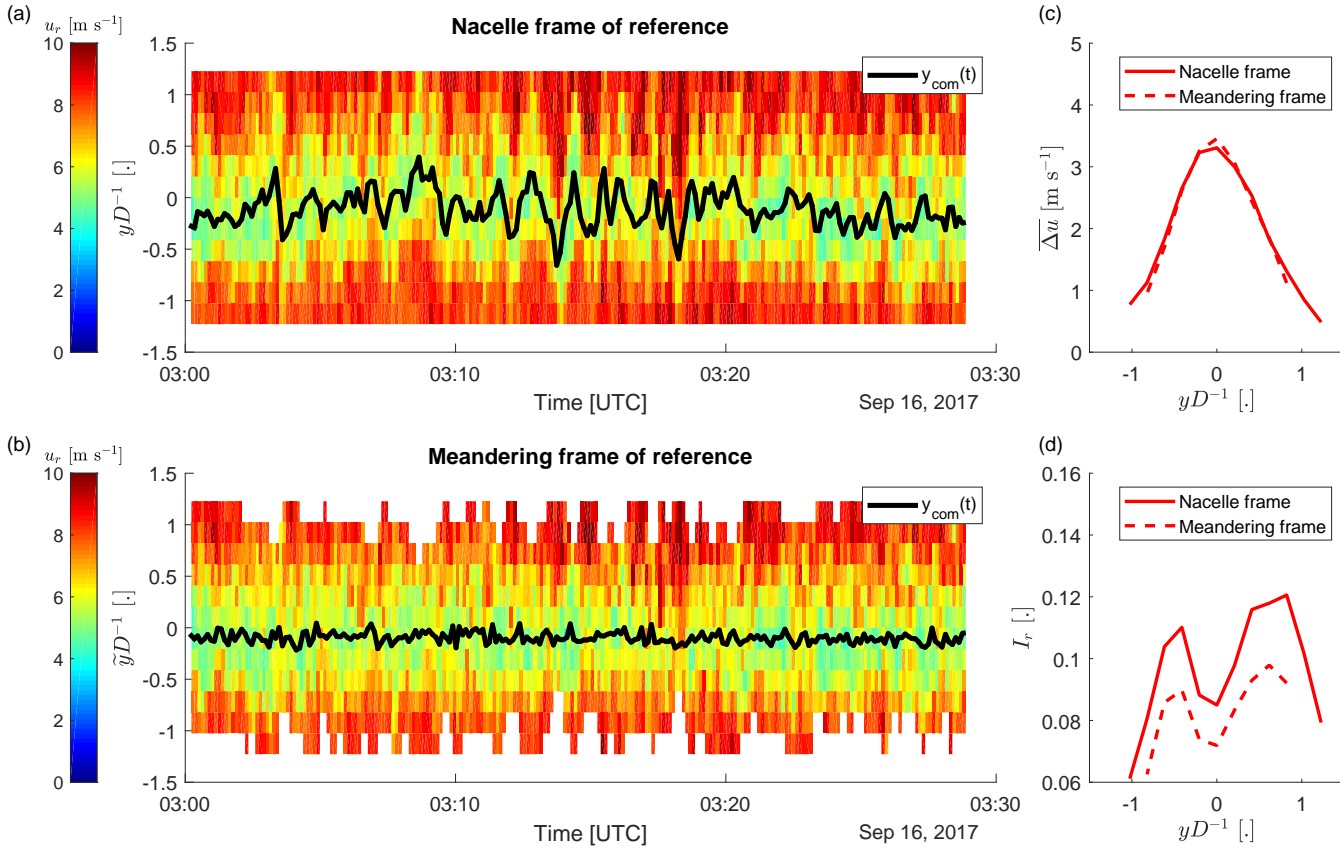

**Figure 11.** Time series of radial velocity profiles at $xD^{-1} = 6$ in the nacelle frame of reference (a) and the meandering frame of reference (b). The corresponding mean longitudinal velocity deficit profiles are shown in (c) and the radial turbulence intensity profiles in (d), respectively.

wake recovery. The transformation method from the nacelle frame of reference to the meandering frame of reference retains
wake meandering on scales smaller than the lateral grid resolution and, therefore, the shown results could be biased towards underestimating the effect of wake meandering on the temporally averaged wake.

### 3.2.2 Turbulence Intensity

The effect of wake meandering on the turbulence intensity across the wake is investigated. An increase of the turbulence intensity from the meandering frame of reference to the nacelle frame of reference can be observed for the example case
illustrating the transformation (Fig. 11d). This effect is quantified with the laterally averaged difference of the turbulence intensity between the two frames of reference. Lateral positions with less than 75% data availability in the meandering frame of reference are discarded.

The results show that the turbulence intensity contributed by wake meandering increases with the strength of wake meandering (Fig. 13a). The magnitude of the turbulence intensity resulting from wake meandering decreases with downstream

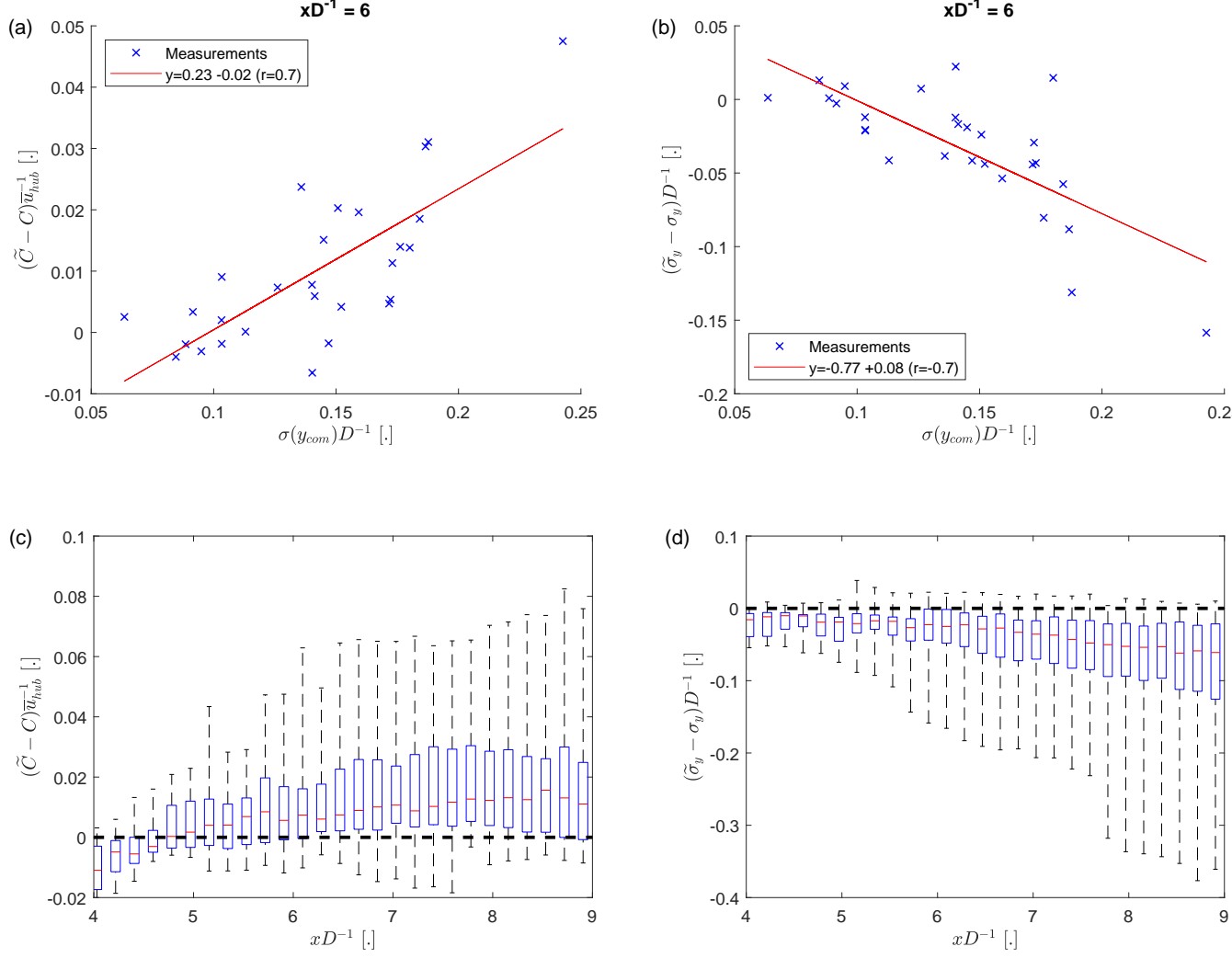

**Figure 12.** Difference of wake depth (a) and wake width (b) between the nacelle frame of reference and the meandering frame of reference as a function the wake meandering strength. Wake depth and wake width of the nacelle frame of reference are $C$ and $\sigma_y$, and of the meandering frame of reference are $\widetilde{C}$ and $\widetilde{\sigma}_y$. Panels (c) and (d) show the distribution of observed differences in depth and width as a function of $xD^{-1}$. The whiskers show the range of the data, the top and bottom of the blue box indicate the 25th and 75th percentile, and the red center marker is the median.

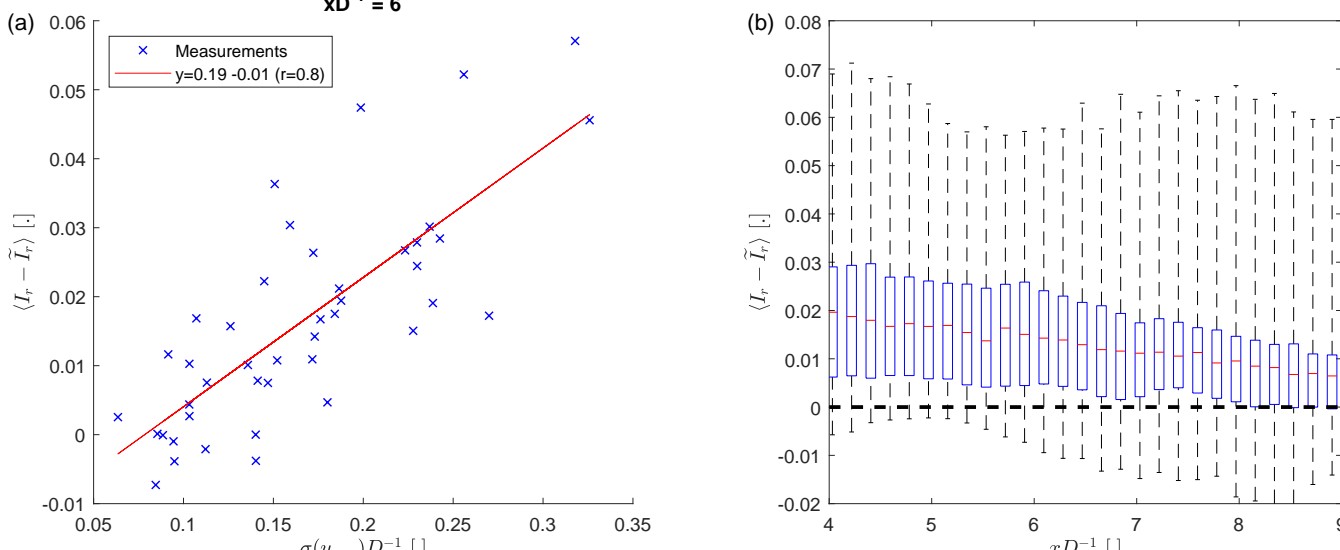

**Figure 13.** Laterally averaged difference of the radial turbulence intensity at $xD^{-1} = 6$ between the nacelle frame of reference ($I_r$) and the meandering frame of reference ($\widetilde{I}_r$) as a function of the wake meandering strength (a). The distribution of observed turbulence intensity differences as a function of $xD^{-1}$ are shown in panel (b) with the whiskers showing the range of the data, the top and bottom of the blue box indicating the 25th and 75th percentile, and the red center marker showing the median. The angle brackets indicate the lateral averaging.

distance (Fig. 13b). This decrease of turbulence intensity due to wake meandering with $xD^{-1}$, despite an increase of the wake meandering strength with $xD^{-1}$ (Fig. 8b), could indicate that the recovery of the velocity deficit plays a more dominant role, compared with the wake meandering strength, in the far wake.

## 4   Summary and Conclusions

Atmospheric field measurements of the wind velocity from two Doppler LiDARs mounted on the nacelle of a utility-scale wind turbine were used to investigate wake meandering. The relationship between the lateral velocity component of the inflow and the instantaneous wake position and the effect of wake meandering on the time-averaged wake were analysed. The main conclusions of this study are:

- In agreement with previous wind tunnel studies, we observe that the strength (or amplitude) of wake meandering increases with the turbulence intensity of the inflow and with downstream distance from the wind turbine. Both trends appear to be linear.

- A correlation between the lateral velocity component and the instantaneous wake position supports the passive advection hypothesis. Further, we found that the quality of the correlation depends on the ratio of the integral time scale of the lateral

velocity to the time delay due to downstream advection, which can be explained by the evolution of the turbulent wind field during the downstream advection.

— Applying a low-pass filter equivalent to $2D$ improves the correlation between $v$ and $y_{com}$ for short downstream distances up to $5D$ supporting the hypothesis that large scale turbulence has an important role in the origin of wake meandering. However, at large downstream distances beyond $5D$ a more suitable low-pass filter threshold is based on the time delay due to downstream advection to remove scales that are expected to be decorrelated due to the evolution of the turbulent wind field.

— The speed at which wake meandering is propagating downstream is smaller than the inflow wind speed at hub height, but larger than the velocity at the wake center. This indicates that the wake is not entirely passive and that at least for the downstream advection process the velocity deficit of the wake itself has an influence. Because the downstream advection time directly affects predictions of the instantaneous wake position, this could introduce a bias towards under estimating the wake meandering strength. The average of inflow wind speed at hub height and the average velocity at the wake center as proposed by Cheng and Porté-Agel (2018) showed reasonable agreement with the observed advection velocities.

— Wake meandering decreases the depth and increases the width of the time-averaged wake and contributes to the turbulence intensity of the wake. For all of those three quantities, the observed effect due to wake meandering was small relative to the base magnitude of the time-averaged wake itself. Only the effect on width of the time-averaged wake increased with downstream distance. The effect on the turbulence intensity decreased with downstream distance and the effect on the wake depth did not show a clear increase or decrease with downstream distance. This suggests that adverse effects on downstream wind turbines resulting from wake meandering become less severe with increasing turbine spacing despite the wake meandering strength increasing with downstream distance.

In the future, the dataset presented here could be used to validate and improve numerical models of wake meandering. These include, for example, the dynamic wake meandering model (Larsen et al., 2008) as well as the statistical wake meandering model of Thøgersen et al. (2017). Incorporating the effect of large scale fluctuations of the wind speed on the downstream advection process and wake meandering is one lead we are planning to pursue in this direction.

*Data availability.* The data is not made available publicly.

## Appendix A: Error of the wake center position due to the lateral velocity

The wake center position computed with Eq. (2) uses the radial velocity instead of the longitudinal velocity. The presence of a lateral velocity component will bias the wake center position, which will be discussed in the following.

To investigate the bias, we assumed an Gaussian wake for the longitudinal velocity component and a constant lateral velocity. The horizontal velocity vector is given by

$$\boldsymbol{u}(y) = \left( u_0 - A \exp\left( -\frac{y^2}{2\sigma_y^2} \right), \, v \right) \tag{A1}$$

with $u_0 = 6 \text{ m s}^{-1}$, $A = 0.4u_0$, $\sigma_y = 40$ m, and a variable lateral velocity $v$. Then the projected line-of-sight velocity that would be measured by the Doppler LiDAR is computed with

$$v_r(y) = \boldsymbol{u} \cdot \boldsymbol{e_r}, \tag{A2}$$

where $\boldsymbol{e_r} = (-\cos(\phi), \sin(\phi))$ is the unit vector in the beam direction for a PPI of the wake scanning Doppler LiDAR (Fig. 2a). The resulting radial velocity profiles are shown in Fig. A1a for $v = 0 \text{ m s}^{-1}$ (solid blue line) and $v = 2 \text{ m s}^{-1}$ (solid black line). Then, the wake center position was computed with Eq. (2) for both cases and they are shown as a vertical dotted lines in Fig. A1a. From the difference of wake center positions between the two cases, a bias of the wake center position is apparent. For the given example in Fig. A1a the difference between the two wake positions is $y_{com}(u_0, v=0) - y_{com}(u_0, v=2) = -11.7$ m.

For each of the 43 cases investigated in Sect. 3 we extracted the maximum lateral velocity observed by Doppler LiDAR in the lateral staring mode and the mean wind speed from SCADA data. We then computed the bias of the wake position analogue to Fig. A1a for each case and compared it with the wake meandering strength in Fig. A1b. The bias of the wake position resulting from the lateral velocity is in all cases smaller than the wake meandering strength. The given biases can be regarded as an upper limit of the error, because we used the absolute maximum of the lateral velocity. Further, it should be noted that the bias of the wake position resulting from the lateral velocity is in the opposite direction to the expected wake displacement with the passive advection hypothesis. Therefore, the error would be reducing the correlations shown in Fig. 5 and not artificially inflating them.

The presence of a vertical velocity will not bias the wake position, because it will affect the radial velocity on both sides of the PPI identically.

## Appendix B: Instantaneous velocity deficit definition

Two definitions of the instantaneous velocity deficit are compared and their effect on the detection of the instantaneous wake position is discussed. The first definition is relative to the mean wind speed of the inflow at hub height given by

$$\Delta u_r(x,y,t) = \overline{u}_{hub} - u_r(x,y,t). \tag{B1}$$

The second definition is relative to the instantaneous ambient wind speed outside of the wake given by

$$\Delta u_r(x,y,t) = \max_y(u_r(x,y,t)) - u_r(x,y,t), \tag{B2}$$

where $\max_y(u_r(x,y,t))$ is the maximum of the radial velocity for a given time and downstream distance, which is taken as the instantaneous velocity outside of the wake. The effect that those two definitions of $\Delta u_r(x,y,t)$ have on the wake position

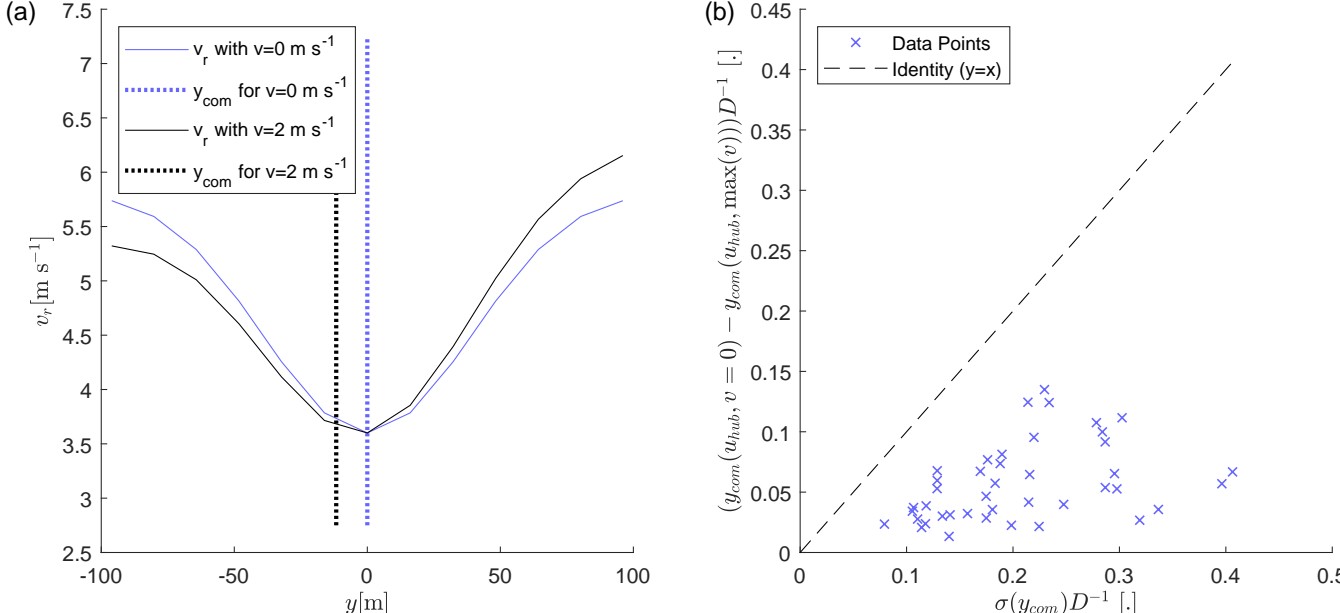

**Figure A1.** The error of the wake center position resulting from the influence of the lateral velocity component. Explanations are in the Text of Appendix A.

computed with Eq. (2) will be investigated by assuming a Gaussian wake given by

$$u_r(y,t) = u_0 - A\exp\left(-\frac{(y - y_0(t))^2}{2\sigma_y^2}\right), \tag{B3}$$

with $A = 3.5$ m s$^{-1}$, $\sigma_y = 40$ m, $y_0(t) = 0.2D\cos(\frac{3\pi t}{1800})$ to model wake meandering, and $u_0$ either having a constant value of 7 m s$^{-1}$ to model a stationary inflow (Fig. B1a) or $u_0 = 7 + \sin(\frac{2\pi t}{1800})$ m s$^{-1}$ to model a non-stationary inflow (Fig. B1d).

Both definitions yield the same velocity deficit for a stationary ambient flow outside of the wake and subsequently identical wake center positions (Fig. B1a-c). For a non-stationary ambient wind speed, the two velocity deficit definitions lead to different results for the wake center position (Fig. B1d-f) and only the definition given by Eq. (B2) reproduces the sinusoidal input for the wake meandering. The above observations from the idealized cases can also be found in our field measurements (Fig. B1g-i). Negative values of $\Delta u$ in Fig. B1f and Fig. B1i were set to zero for the computation of $y_{com}$, because the center-of-mass

method (Eq. 2) does not work for negative values.

We will use the velocity deficit definition based on $\max_y(u_r(x,y,t))$ in this paper, because it provides a better detection of the wake center position. However, the results presented in Sect. 3 hold for both definitions of the velocity deficit. It should be noted that the definition based on $\max_y(u_r(x,y,t))$ removes a portion of the inflow turbulence from the velocity deficit field of the wake and, therefore, the turbulence intensity should be computed from $u_r(x,y,t)$ directly and not $\Delta u_r(x,y,t)$.

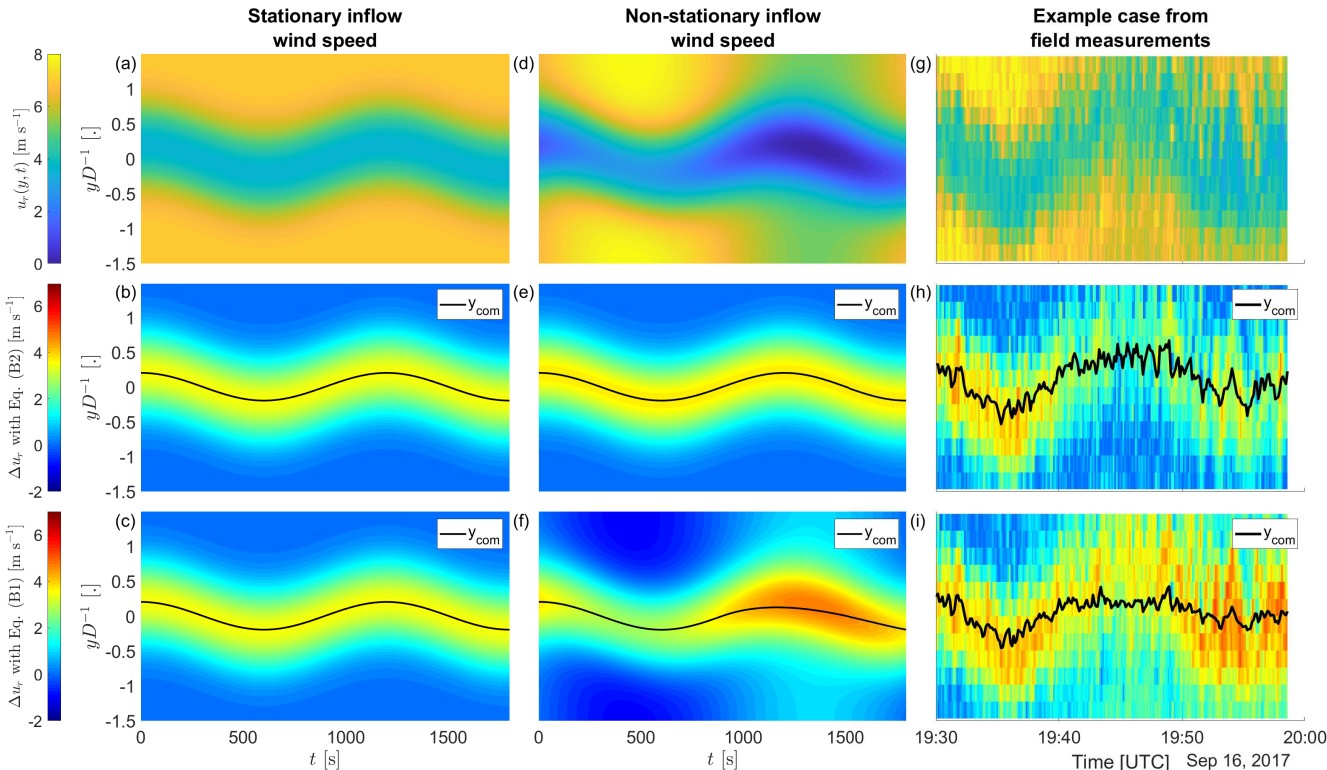

**Figure B1.** Time series of velocity profiles (top row) and derived velocity deficits with Eq. (B2) based on $\max_y(u_r(x,y,t))$ (middle row) and with Eq. (B1) based on $\overline{u}_{hub}$ (bottom row). The left two columns present a Gaussian wake with sinusoidal wake meandering for a constant inflow wind speed (left column) and a non-stationary inflow wind speed (middle column). The right column shows an example case from the data set presented in Sect. 3, which was selected for its pronounced differences between the two velocity deficit definitions.

*Author contributions.* P.B. contributed to the data curation, formal analysis, conceptualization, methodology, software, validation, visualization, and writing (original draft). F.P.-A. contributed to the conceptualization, funding acquisition, project administration, supervision, and writing (review and editing). C.D.M contributed to the funding acquisition, resources, data curation, and investigation.

*Competing interests.* The authors declare that they have no conflict of interest.

*Acknowledgements.* The authors would like to thank Kirkwood Community College for their cooperation and allowing access to their wind
turbine. We also extend our appreciation to Clipper Windpower for granting access to technical data on the Liberty wind turbine. This research
was funded by the Swiss National Science Foundation (grant number: 200021_172538), the Swiss Federal Office of Energy (grant number:

SI/502135-01), the National Science Foundation Iowa EPSCoR (Grant No 1101284), and the Center for Global and Regional Environmental Research (CGRER), University of Iowa.

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
