# Peer review of "Field measurements of wake meandering at a utility-scale wind turbine with nacelle-mounted Doppler LiDARs"

_Wind Energy Science, 2021_

## Author Response (AR1)

**General Comments**

This article is presenting the results of a field study of the properties of the wake generated by a full-scale wind turbine. The study is based on wind observations acquired by two scanning wind lidars installed on the nacelle of the wind turbine. Flow observations previously acquired mainly in wind tunnel studies now with the help of long-range scanning wind lidars can be also performed around full-scale wind turbines. The measurement configuration enabled the study of the mean and standard deviation of the wind speed in the wake, as well as its meandering in relation to the fluctuations of the transverse wind component. The study of these parameters is important for enhancing our understanding of the interaction between wakes in a wind farm. The study is well structured and written, and the methods used along with the corresponding results are well described.

We are grateful to the referee for the comments, which helped to improve the manuscript. Our replies to the comments below are shown in blue below. The line numbers given in the reply refer to the revised manuscript. A manuscript with tracked-changes is also provided.

1.  The wind observations used in this study have been acquired up- and downwind of wind turbine that is located in an urban area. From the scale presented in Figure 1, one can deduct that in the vicinity of the wind turbine and there are buildings. For certain wind directions those buildings are within then measuring range of the scanning wind lidar measuring downwind. I think that it will be helpful for the understanding and the interpretation of the results presented in this study, to discuss if the characteristics of the topography are expected to have an impact on the data which were selected for this analysis.

    In the following, we show that the heterogeneity and topography of the surface around the wind turbine should not interfere with the measurements at hub height by (i) clustering our results for different wind direction, and (ii) based on the blending height concept.

    First, we investigated the effect of individual buildings in the vicinity of the wind turbine by clustering the measurement data according to the wind direction (see Fig. 1 below). If individual buildings or topography features have a pronounced influence on the results, then the data from a particular wind direction cluster should exhibit marked differences to the other two clusters. The below Fig. 2 shows that the wind direction clusters are mixed throughout the main results of the manuscript, indicating that individual roughness elements do not have a strong impact on the results.

    The roughness sublayer is the layer directly above the surface where individual surface roughness elements induce horizontal variability of the flow statistics. We assume that if we are measuring above the roughness sublayer, then the footprints of individual buildings have been blended (or spatially averaged) by the turbulence and should not interfere with the measurements. This is also often referred to as the blending height concept in literature. While the depth of the roughness sublayer depends on many factors and is not fully understood yet (Mahrt, 2000), we provide two practical approaches below:
    - Turbulent flux measurements with the eddy-covariance method need to be conducted above the roughness layer to measure a spatially averaged signal representative of the local area (e.g. Aubinet et al., 2012). The range given in literature for the roughness sublayer height in urban environments ranges from $1.5h_c$ over densely built-up areas to $5h_c$ over low-density areas (Grimmmond and Oke, 1999). Assuming a building height of $h_c = 10$ m for the area around the wind turbine (two and three-story buildings), the hub height of the wind turbine is above the roughness sublayer for both

build-up densities. An example with eddy covariance measurements that were conducted at three times the building height above a city and are assumed to be in the constant flux layer is shown in Velasco (2009).

- Raupach (1994) estimates the depth of the roughness sublayer above plant canopies starting at the displacement height ($d$) with $z^* = 2 * (h_c - d)$, with $d$ between $\frac{2}{3}h_c$ and $\frac{1}{30}h_c$. Therefore, the small patches of wood are also not expected to have any effect on the wake measurements.

In summary, we believe our measurements at the hub height (77 m) are above the roughness sublayer and hence we would not expect that the footprints of individual buildings affect the measurements based on the blending height concept.

We added a short paragraph at beginning of Sect. 3 (lines 149 to 153) that states that the results are not differentiated according to the wind direction following scale arguments and a cluster analysis without going into the details shown here.

*References*

- *Mahrt, L. Surface Heterogeneity and Vertical Structure of the Boundary Layer. Boundary-Layer Meteorology 96, 33–62 (2000). https://doi.org/10.1023/A:1002482332477.*
- *Aubinet, M., Vesala, T., Papale, D., 2012. Eddy Covariance: A Practical Guide to Measurement and Data Analysis. Springer, 460 pp.*
- *Grimmond C.S.B., Oke T.R. (1999) Aerodynamic properties of urban areas derived from analysis of urban form. J Appl Meteorol 38:1262–1292*
- *Velasco, E., Pressley, S., Grivicke, R., Allwine, E., Coons, T., Foster, W., Jobson, B. T., Westberg, H., Ramos, R., Hernández, F., Molina, L. T., and Lamb, B.: Eddy covariance flux measurements of pollutant gases in urban Mexico City, Atmos. Chem. Phys., 9, 7325–7342, https://doi.org/10.5194/acp-9-7325-2009, 2009.*
- *Raupach, M. R.: 1994, Simplified Expressions for Vegetation Roughness Length and Zero-Plane Displacement as Functions of Canopy Height and Area Index, Boundary-Layer Meteorol.71, 211–216.*

[Figure]

*Figure 1: Histogram of the wind direction from the SCADA data for the data sets analyzed in the manuscript. The colors show the three wind direction clusters defined to investigate the effect of surface heterogeneity on the results.*

[Figure]

*Figure 2: The main results from the manuscript are shown for different wind direction clusters. The panels (a) to (c) correspond to the results of Section 3.1 (namely Fig. 4a, Fig. 7a, and Fig. 9). The panel (d) to (f) corresponds to the results of Section 3.2 (namely Fig. 11a, Fig. 11b, and Fig. 12a).*

2. Equation 1 presents the way that the authors estimate the instantaneous longitudinal component of the wind speed vector. This equation requires the assumption that the instantaneous transverse and vertical (in the case when the PPI scan is not horizontal) wind components are zero. I suggest that the authors should state that in the manuscript and discuss about the validity of this assumption. Furthermore, equation 1 implies that the wind turbine was all the time aligned to the mean wind direction. It would be interesting to present some data that support this. An idea is to see if the mean lateral component is zero over the periods examined in this study.

We abandoned the assumption of a zero lateral velocity and removed it from the manuscript. We now use the radial velocity as measured by the Doppler LiDAR directly for the computation of the wake center position and the turbulence intensity. The following changes have been made to the manuscript in response to this comment:

- Eq. (1) from the previous version of the manuscript has been removed as it is no longer used. Section 2.2 and has been revised to use the radial velocity form the Doppler LiDAR for the computation of the wake center position.
- The appendix A1 has been added to the manuscript showing that the error of the wake center position that is caused by the lateral velocity or the vertical velocity is not invalidating the results.
- We update Figures 5 to 10, and 13 with the new results.
- Longitudinal velocities have been changed to the radial velocity throughout the manuscript text were appropriate.
- Section 3.2.1 continues to use longitudinal velocities, because the assumption of zero lateral velocities holds for the temporally averaged wake. We added a conversion of radial velocities to longitudinal velocities to the text of this section (lines 228-232).

The alignment of the wind turbine with the wind direction is to some extend ensured with the check that the wake is within the field-of-view of the Doppler LiDAR (lines 142-143). The mean yaw misalignment of the wind turbine is also available from the SCADA data (shown in Fig. 3 below) and it is in all but one case within 2°. Accounting for the yaw misalignment with $1/\cos\gamma$ in Eq. (1) has no discernable effect on the results (and was not added to the manuscript).

[Figure]

*Figure 3: Mean yaw misalignment of the wind turbine from the SCADA data for all investigated wake scans.*

**Specific Comments**

1. Lines 60 – 65: I have a few questions regarding the measuring configuration of the downwind looking scanning wind lidar. I think that it would be very useful for the comprehension of the study if the authors clarify the following:

   1. Was the scanner head moving continuously during one PPI scan or was it still for each azimuth step?

      The scanner head was moving continuously with a speed of 6° per second. We added this information to the manuscript (lines 61-63) and added a new panel to Figure 2 in the manuscript that illustrates the scanner movement. The scanner speed together with the measurement frequency of 3 Hz resulted in azimuth resolution of 2° given in the manuscript. The acceleration phase of the scanner head at az=168° is negligible, but the arrival of the scanner head at az=192° did not always exactly line up with the 13th measurement (sometimes it was at 191.8° at the time of the 13th measurement). The reason for this is that the measurement frequency has small fluctuations and it is likely that the same is true for the scanner movement from PPI to PPI (added in lines (75-77). In those cases the 14th measurement is used.

   2. It is stated that 230 PPI scans were acquired over 29 minutes. Based on the time completion of one PPI scan (7.2 seconds), the 230 PPI scans should be completed in 27.6 minutes. Why is there a discrepancy?

      The time discrepancy comes in part from rounding the mean duration of 28 minutes and 40 seconds up to 29 minutes. And in part from the short resting time of the scanner at the turn-around points at 168° and 192°, which has not a consistent length. The new panel to Fig. 2 in the manuscript shows an example for the inconsistency of resting times (compare first and second turn-around at 168°). As a result the time a PPI took is not consistent and can be as long as 7.6 seconds. However, we accounted for those inaccuracies in the post-processing, because we use the actual starting times

of the PPIs as our timestamps, when computing the cross-correlation with lateral velocity (clarified in lines 75).

3.  Does the scanning direction of the PPI scan stay the same or does it alternate between consecutive scans?

    The scanning direction stayed the same. We added a second panel to Figure 2 in the manuscript, which shows the scanner path over time.

4.  What was the tilt angle of the line-of-sight measurements during the PPI scan? Is it expected a tilt of the whole nacelle when the wind turbine was operating? Why have the wake scans been scheduled every second hour?

    The tower tilt should not affect the results following the below arguments:
    - If the tower tilts with increasing loads, the PPI scan would intersect the wake below hub height. However, the wake position is not affected by this assuming a circular shape of the wake. Even in the presence of wind veer that leads to an elliptical shape of the wake, this would only lead to an offset of the wake position that does not affect the results. Therefore, a tilting of the tower and the resulting beam misalignment should affect the results.
    - We used values of the tower top displacement for nodding oscillations found in literature to estimate the tower tilt. These values are based on modeling results for conditions above the rated wind speed. A maximum tower top displacement ($\Delta x$) of 0.2 m was given in Bossanyi (2003). Two further estimates found in grey literature provided similar values ($\Delta x = 0.2$ m in a technical report by Hooft et al. (2003) for a turbine with a hub height of 92 m and $\Delta x = 0.12$ m by Mate Jelavic et al. (2007) in conference proceedings). To estimate the effect on the lidar beam, we assume that the tower is stiff and computed the beam misalignment with $\tan^{-1}(\Delta x/h_{hub})$, which results in an beam misalignment from the horizontal of 0.12° for $\Delta x = 0.2$ m. Assuming that wake center remains at hub height with downstream propagation, then the PPI would cut the wake 1.8 m below hub height at $xD^{-1} = 9$.

    Other scan patterns occupied the remaining 1.5 hours (a 30-minute planar scan with a higher spatial resolution and larger azimuth range used by Fuertes et al. (2018), a 30-minute volumetric scan used by Brugger et al. (2019), and a 30-minute stare along the rotor axis).

    *References*

    *Bossanyi, E.A. (2003): Wind Turbine Control for Load Reduction. Wind Energy, 6:229-244. DOI: 10.1002/we.95.*

    *Carbajo Fuertes, F.; Markfort, C.D.; Porté-Agel, F. Wind Turbine Wake Characterization with Nacelle-Mounted Wind Lidars for Analytical Wake Model Validation. Remote Sens. 2018, 10, 668. https://doi.org/10.3390/rs10050668*

    *Brugger, P.; Fuertes, F.C.; Vahidzadeh, M.; Markfort, C.D.; Porté-Agel, F. Characterization of Wind Turbine Wakes with Nacelle-Mounted Doppler LiDARs and Model Validation in the Presence of Wind Veer. Remote Sens. 2019, 11, 2247. https://doi.org/10.3390/rs11192247.*

5. One wind lidar was measuring the wake with scans that lasted 29 seconds while the second wind lidar was measuring the transverse wind component for 14 seconds. Why did the authors select a different period for the two measuring modes?

   After measuring the transverse velocity component for 14 minutes, the upstream facing LiDAR changed to another azimuth position (az=0° or parallel to the rotor axis) and was measuring the longitudinal velocity component upstream of the wind turbine for the remaining 14 minutes (with a one minute gap in between). However, our plans for those scans did not work out and they were not included to the manuscript for that reason.

5. Lines 90-92: The authors state "that the range gates closer than y = 117 m showed time series that were inconsistent with the flow behaviour observed at further distances". What kind of inconsistencies were observed?

   The first four range gates of this Doppler LiDAR type have generally bad measurements (independent from the setup of the LiDAR). For the fifth and sixth range gate, which have reliable observations typically, we observed velocities biased towards positive values and higher standard deviations compared to range gates at greater distances (see Fig. 4 below for an example). We believe the issues of the fifth and sixth range gate are caused by the influence of the wind turbine on the flow field that might extend beyond the rotor diameter (e.g. a flow displacement due to a blockage effect and the resulting turbulence). We added a clarification to the manuscript in lines 96-98.

[Figure]

Figure 4: Standard deviation of the lateral velocity component (left panel) and time series the radial velocity from the Doppler LiDAR in the lateral staring mode. The black dot (or black dashed line) indicate the 7th range gate at y=117 m. This figure uses raw data from the Doppler LiDAR without any quality control to illustrate error. The bad measurements of the first four range gates are visually apparent. A bias towards positive velocities (more red) can be seen for the 5th range gate (for other cases as well for the 6th range gate). The increase of the standard deviation at $y > 1600$ m is caused by noise due to a decreasing SNR with distance.

6. Line 96: Over which periods were the standard deviation of the lateral component and the integral time scale computed? Was it over a 14-minute period?

   Yes, it was the 14-minute period. We added a clarification to the text (line 106)

7. Line 110: I am not sure if I understand correctly equation 6. The term Δx/u_hub corresponds to the advection time between two measuring locations along the x-axis. Is the u_hub the instantaneous measurement from the nacelle anemometer or a mean and is there an expected flow distortion in these measurements due to the presence of both the nacelle and of the two scanning wind lidars? Why is the u_hub and not the u_max[x] used here? And why does the advection time have the subscript "hub". Maybe I have misunderstood this point, so I would really appreciate if you can clarify it.

We modified Sect. 2.4 and Sect. 3.1.1, because the previous approach was not very intuitive in retrospect.

- In the previous version of the manuscript, we divide the data set into two parts: One part with a successful detection of the advection velocity with the cross-correlation, for which we used advection velocity and corresponding time delay with the subscript "adv". A second part where the detection of the advection velocity was not successful, and for we used the mean wind speed given by $\bar{u}_{hub}$, which was indicated subscript "hub" for the time delay.
- This approach has been abandoned with revised manuscript and Sect. 2.4 only introduces the advection velocity $u_{adv}$, now (Eq. 4 in the manuscript). Following that, the revised Sect. 3.1.1 and Fig. 5 therein provide the time delay directly in the legend with the corresponding equation. Data points with a time delay based on the mean wind speed are shown for all cases now.

Due to those changes, the equation was removed from the manuscript, because it is no longer needed. Nevertheless, below are the answers to the specific questions:

- $\bar{u}_{hub}$ is the temporal mean value over the duration of the wake scan from the nacelle anemometer. A clarification was added to the text in line 54 and we use bars to indicate temporal averages throughout the manuscript, now.
- A flow distortion affecting $\bar{u}_{hub}$ is investigated by comparing it to independent measurements of the mean wind speed. First, we compare $\bar{u}_{hub}$ with the measurements of a cup anemometer on the 80 m boom of a nearby meteorological tower (the meteorological tower is described in detail in Vahidzadeh and Markfort, 2019). Second, we compared $\bar{u}_{hub}$ with the radial velocity measured by the upstream facing Doppler LiDAR, when it was in a longitudinal staring mode parallel to the rotor axis (Fig. 5 below, left panel). These longitudinal stares have not been introduced in the manuscript or used in any other publication yet. From the comparison with the independent measurements, it can be seen that $\bar{u}_{hub}$ compares well overall with a small bias towards higher wind speed, which might be explained by a flow acceleration around the nacelle.
- We use $\bar{u}_{hub}$ here, because a downstream advection with the mean wind speed is a common assumption in the literature for wake meandering and, therefore, is a good choice for the comparison with the advection velocities we determine with the cross-correlation approach. The $\max(u_r(x,t))$ introduced in the previous section transfers the concept of the velocity deficit from the mean flow field to the instantaneous flow field (if we used $\bar{u}_{hub}$ for this purpose, then we would get negative velocity deficits).

*References*

*Vahidzadeh, M.; Markfort, C.D. Modified Power Curves for Prediction of Power Output of Wind Farms. Energies **2019**, 12, 1805. https://doi.org/10.3390/en12091805*

[Figure]

*Figure 5: Comparison of $\overline{u}_{hub}$ from the nacelle mounted anemometer with a cup anemometer at 80 m above ground level at a nearby meteorological tower (left panel) and with the upstream facing Doppler LiDAR in a staring mode parallel to the rotor axis at a range gate 243 m upstream of the wind turbine (right panel).*

8. Line 125: Can you please clarify how Sec. 3.1.2 justifies u_adv=u_hub

   Section 3.1.2 shows that the observed advection velocities are in all cases smaller or equal to $\overline{u}_{hub}$. Because the error grows with $u_{adv}$, assuming $u_{adv} = \overline{u}_{hub}$ will overestimate the error in most cases and, therefore, be on save side. A clarification was added to the text in line 131.

9. Line 137: Is it larger velocity deficit or smaller velocity deficit? A wake that is partially outside the field of view of the lidar should not result in an observed smaller velocity deficit?

   The velocity deficit at an outside grid point will increase if a wake, that is initially centered within the LiDARs field-of-view, moves towards the outside over time. We rephrased the sentence and use mathematical expressions instead of words for clarity (lines 142-143). This check was designed and verified to be triggered by cases with a misaligned wind turbine, or by cases with a very high turbulence intensity, when the wake meandering was so strong that the azimuth range of the scans became too small to capture it.

10. Lines 139-140: It would be interesting to see what is the distribution of wind speed and turbulence intensity of these 35 wake scans. Also, what was the corresponding yaw direction?

    The below figure shows the distribution of wind speed and turbulence intensity and their corresponding nacelle positions (a distribution of the yaw misalignment was shown in our reply to the second major comment above). We did include this information to manuscript in lines 145-148 and Fig. 4. The manuscript stated longitudinal turbulence intensity here previously, which has been corrected to the lateral turbulence intensity.

[Figure]

*Figure 6: Distribution of wind speed (top left) and lateral turbulence intensity (top right) and the corresponding nacelle positions (bottom).*

11. Line 152: Can you please clarify what is the Ti,u ?

$T_{i,v}$ is the integral time scale of the lateral velocity component (we added a clarification to lines 167-168). We also renamed the lateral turbulence intensity from $TI_v$ to $I_v$, because it was too similar with $T_{i,v}$ and might have been confused with each other.

12. Page 9, Figure 4: The number of data points in Figures 4 a, b and c shouldn't be equal to 35, one for each of the selected cases? And also, why is the number of blue crosses different from the black ones?

In the previous version of the manuscript the sum of blue and black data points together was equal to 35. We used the advection velocity based on the cross-correlation if available (black), and for the remaining data points we used mean wind speed as advection velocity (blue).

This was not a very intuitive approach and we changed Fig. 5a-c to show the data points based on the mean wind speed for all cases (black) and in addition, the data points based on the cross-correlation for the available cases (blue).

As a side note here, we noticed a mistake in our previous selection of the suitable cases: the time stamps of the SCADA data were sometimes one second to early (e.g. hh:59:59 instead of hh+1:00:00). This led to some wake scans being discarded wrongly, because we believed the SCADA data was missing and after adjusting the post-processing for this issue, we have now

43 suitable wake scans. Further, the outlier mentioned in Fig. 8 of the previous version of the manuscript is no longer an issue with this change (here both time stamps at the beginning and the end were one second to early).

13. Page 14, Figure 9: If I counted correctly 23 values appear in the scatter plot. Are some of the 35 selected data sets filtered out? Also, what does the dashed line represent? Is it an identity line or a fit?

    Yes, only wake scans with a successful detection of the advection velocity are shown in Fig. 10. The rejection of wake scans with an overall correlation smaller than 0.5 between $v$ and $y_{com}$, or no local maximum of the cross-correlation in the search space is the reason for the missing data points (described in lines 126-127 of the manuscript). The dashed line is the identity and we added a legend to Figure 9 clarifying that.

14. Line 255: The authors write "for short downstream distances". I think that this point will be clearly if they quantify those distances in terms of rotor diameters.

    We quantified "short downstream distances" with up to $5D$ for clarification (lines 275-277).

**Technical Corrections**

1. I suggest to in general replace the word "stares" that is used to describe the operational mode of the forward wind lidar with the "staring mode"

   We rephrased the term as suggested at all instances in the manuscript.

2. Line 26: Please correct the "hypotheis" with "hypothesis"

   Corrected (line 26).

3. Line 27: Please correct the "hypotheis" with "hypothesis"

   Corrected (line 27).

4. Line 27: The authors write: "The passive advection hypothesis also forms the basis of the dynamic wake meandering model …" why the use the word "also" in this sentence?

   We removed "also" (line 27).

5. Line 33: Maybe the verbs "described" or "reported" are more suitable than the word "established"

   We replaced "established" with "reported" (line 33).

6. Lines 36 – 39: I think that the point of the two sentences is the same. I suggest avoiding the repetition

   We agree and removed the second sentence as the spectral aspect is already included in the phrase "for large wavelengths" in the first sentence (line 38).

7. Line 50: I suggest re-writing the sentence "Measurement were conducted" with "the measurement campaign was conducted…" or something similar.

   We rephrased as suggested (line 49).

8.  Line 65: I suggest replacing the "used a horizontal state at a 90o angle" with "was measuring in a horizontal staring mode at a 90o angle

    Rephrased as suggested (line 67).

9.  Line 91: Replace "farther" with "futher"

    We replaced "farther" with "greater" to make the sentence unambiguous (line 98).

10. Line 94: Replace the "latereal" with "lateral"

    Corrected (line 100).

11. Line 110: Please correct the beginning of the sentence "The with a moving …"

    We corrected the sentence by removing ". The" (line 119).

12. Line 190: Replace the "synchronisations" with "synchronisation"

    Corrected (line 207).

13. Figure 8 – Label: Replace the "coss-correlation" with "cross-correlation"

    Corrected (label of Fig. 9).

14. References: In some references the DOI is not presented correctly, e.g. the reference of Vermeer et al. 2003 and Sanderse et al. 2011. Furthermore, the Journal names should be abbreviated according to the Journal Title Abbreviations by Caltech Library

    We implemented journal abbreviations and removed the doubled "https://doi.org/" before DOI-numbers in the references (lines 355 onwards).

We are grateful to the referee for the comments, which helped to improve the manuscript. Our replies to the comments below are shown in blue below. The line numbers given in the reply refer to the revised manuscript. A manuscript with tracked-changes is also provided.

**Major issue**

The data is analysed without any wind sector information. The surrounding of the turbine has different roughness lengths and obstacles that can cause different internal boundary layers effecting the short distance behind the turbine, measured free wind and wind speed at hub height. Things around the turbine (Financial Park, Johns Hall etc.) look like around 9-10m tall. It is too optimistic to assume that they do not have impact. Therefore, selection criteria are not clear to me. Compared (or summed up) results might be from different conditions and current filtering methods might not be enough. Some inconsistent data (see minor issues) might be product of this situation and that should be addressed at least at discussions or conclusion. If it is assumed that the wind direction does not matter for the analysis, a proof comparison from different sectors with similar conditions can be used.

In the following, we show that the heterogeneity and topography of the surface around the wind turbine should not interfere with the measurements at hub height by (i) clustering our results for different wind direction, and (ii) based on the blending height concept.

First, we investigated the effect of individual buildings in the vicinity of the wind turbine by clustering the measurement data according to the wind direction (see Fig. 1 below). If individual buildings or topography features have a pronounced influence on the results, then the data from a particular wind direction cluster should exhibit marked differences to the other two clusters. The below Fig. 2 shows that the wind direction clusters are mixed throughout the main results of the manuscript, indicating that individual roughness elements do not have a strong impact on the results.

The roughness sublayer is the layer directly above the surface where individual surface roughness elements induce horizontal variability of the flow statistics. We assume that if we are measuring above the roughness sublayer, then the footprints of individual buildings have been blended (or spatially averaged) by the turbulence and should not interfere with the measurements. This is also often referred to as the blending height concept in literature. While the depth of the roughness sublayer depends on many factors and is not fully understood yet (Mahrt, 2000), we provide two practical approaches below:

- Turbulent flux measurements with the eddy-covariance method need to be conducted above the roughness layer to measure a spatially averaged signal representative of the local area (e.g. Aubinet et al., 2012). The range given in literature for the roughness sublayer height in urban environments ranges from $1.5h_c$ over densely built-up areas to $5h_c$ over low-density areas (Grimmmond and Oke, 1999). Assuming a building height of $h_c = 10$ m for the area around the wind turbine (two and three-story buildings), the hub height of the wind turbine is above the roughness sublayer for both build-up densities. An example with eddy covariance measurements that were conducted at three times the building height above a city and are assumed to be in the constant flux layer is shown in Velasco (2009).
- Raupach (1994) estimates the depth of the roughness sublayer above plant canopies starting at the displacement height ($d$) with $z^* = 2 * (h_c - d)$, with $d$ between $\frac{2}{3}h_c$ and $\frac{1}{30}h_c$. Therefore, the small patches of wood are also not expected to have any effect on the wake measurements.

In summary, we believe our measurements at the hub height (77 m) are above the roughness sublayer and hence we would not expect that the footprints of individual buildings affect the measurements based on the blending height concept.

We added a short paragraph at beginning of Sect. 3 (lines 149 to 153) that states that the results are not differentiated according to the wind direction following scale arguments and a cluster analysis without going into the details shown here.

*References*
- *Mahrt, L. Surface Heterogeneity and Vertical Structure of the Boundary Layer. Boundary-Layer Meteorology 96, 33–62 (2000). https://doi.org/10.1023/A:1002482332477.*
- *Aubinet, M., Vesala, T., Papale, D., 2012. Eddy Covariance: A Practical Guide to Mea-surement and Data Analysis. Springer, 460 pp.*
- *Grimmond C.S.B., Oke T.R. (1999) Aerodynamic properties of urban areas derived from analysis of urban form. J Appl Meteorol 38:1262–1292*
- *Velasco, E., Pressley, S., Grivicke, R., Allwine, E., Coons, T., Foster, W., Jobson, B. T., Westberg, H., Ramos, R., Hernández, F., Molina, L. T., and Lamb, B.: Eddy covariance flux measurements of pollutant gases in urban Mexico City, Atmos. Chem. Phys., 9, 7325–7342, https://doi.org/10.5194/acp-9-7325-2009, 2009.*
- *Raupach, M. R.: 1994, Simplified Expressions for Vegetation Roughness Length and Zero-Plane Displacement as Functions of Canopy Height and Area Index, Boundary-Layer Meteorol.71, 211–216.*

[Figure]

*Figure 1: Histogram of the wind direction from the SCADA data for the data sets analyzed in the manuscript. The colors show the three wind direction clusters defined to investigate the effect of surface heterogeneity on the results.*

[Figure]

*Figure 2: The main results from the manuscript are shown for different wind direction clusters. The panels (a) to (c) correspond to the results of Section 3.1 (namely Fig. 4a, Fig. 7a, and Fig. 9). The panel (d) to (f) corresponds to the results of Section 3.2 (namely Fig. 11a, Fig. 11b, and Fig. 12a).*

**Minor issues**

Page 2/line 50: Although I have found the location quite easily over Google Earth, I think you should add the exact coordinates (41.916578°, -91.650871°) to the paper.

We added "The wind turbines coordinates are 41.9165° latitude and -91.6508° longitude." to the caption of Figure 1.

Page 3/Lines 65: Says "The rejection criteria for wake scans not suited for further analysis based on data quality, turbine yaw activity, and inflow characteristics will be presented at the beginning of Sect. 3." But Section 3 does not give any information about directional rejection (if any). If all wind directions are accepted, wouldn't there be a discussion about the wind flow coming from urban areas? Wouldn't that effect the lateral or vertical advection?

The rejection criteria did indeed not include any directional rejection, because we assume that the effects of individual roughness elements have been spatially averaged according to the blending height concept at hub height. A verification of this assumption has been presented in our reply to the major issue above with the directional clustering. A paragraph stating this has been added to the manuscript (lines 149-153).

As a side note here, we noticed a mistake in our previous selection of the suitable cases: the time stamps of the SCADA data were sometimes one second to early (e.g. hh:59:59 instead of hh+1:00:00). This led to some wake scans being discarded wrongly, because we believed the SCADA data was missing and after adjusting the post-processing for this issue, we have now 43 suitable wake scans. Further, the outlier mentioned in Fig. 8 of the previous version of the manuscript is no longer an issue with this change (here both time stamps at the beginning and the end were one second to early).

Page 4/Figure 2 I did not understand the parenthesis saying, not to scale. (Sorry)

The rectangle representing the nacelle and thickness of the line indicating the rotor are too big given the x-axis and y-axis of the plot. If we plotted them according to the true dimension of the wind turbine, they would be too small to be useful.

Page 4/ Line 74 I think radial velocities are converted to vertical components in x-axes is rather more correct then are corrected.

The sentence has been removed from the manuscript and we now use the radial velocities directly for the analysis following a comment from referee #1 (page 5, lines 82-84).

Page 4/ Line 76 I don't understand the simplification of the trigonometric equation! Why do you need to do that?

This approximation has been introduced for an easier to understand presentation of the results. The cross-sections of the Doppler LiDAR scans are a segment of a circle (see Figure 2 in the manuscript) and the correct presentation in the various figures of the manuscript would be a radial distance with an azimuth angle as the ordinate. However, by assuming $x = r \cos^{-1} \phi \approx r$ we can label a particular cross-section with a single downstream distance, while using an ordinate in the units of meter. We believe an ordinate in meter will be more familiar to most readers and it can be directly compared with the rotor diameter.

Page 5/line 83 An example figure would be nice to see the steps 5 and 6 to see and understand the quality of the process since no single instantaneous wake measurement is shared.

Examples of instantaneous wake and the resulting wake center positions are shown in the results section. Fig. 6a and 6b, Fig. 7, and Fig. 11a show instantaneous wake measurements and the instantaneous wake center position from Eq. (2) is also shown in those figures.

Page 5 / Line 91 See my comments about Page 3/line 65 for inconsistent data

The first four range gates of this Doppler LiDAR type have generally bad measurements (independent from the setup of the LiDAR). For the fifth and sixth range gate, which have reliable observations typically, we observed velocities biased towards positive values and higher standard deviations compared to range gates at greater distances (see Fig. 3 below for an example). We believe the issues of the fifth and sixth range gate are caused by the influence of the wind turbine on the flow field that might extend beyond the rotor diameter (e.g. a flow displacement due to a blockage effect and the resulting turbulence). However, beyond the 7th range gate until the signal-to-noise ratio of starts to decrease at 1600 m, the turbulence is fairly homogenous. We added a more precise description of this issue and our reasons to choose the 7th range gate for the lateral velocity to the manuscript (page 5, lines 96-99).

[Figure]

Figure 3 : Standard deviation of the lateral velocity component (left panel) and time series the radial velocity from the Doppler LiDAR in the lateral staring mode. The black dot (or black dashed line) indicate the 7th range gate at y=117 m. This figure uses raw data from the Doppler LiDAR without any quality control to illustrate error. The bad measurements of the first four range gates are visually apparent. A bias towards positive velocities (more red) can be seen for the 5th range gate (for other cases as well for the 6th range gate). The increase of the standard deviation at y>1600 m is caused by noise due to a decreasing SNR with distance.